# Outward foreign direct investment and GVC position of manufacturing industry: A perspective on China's general trade and processing trade structure

Fei Ren[1], Dong Le[2], Ziyu Hu[1]*

1 School of Economics, Zhongnan University of Economics and Law, Wuhan, Hubei, China, 2 School of Business, Wenzhou University, Wenzhou, Zhejiang, China

* 994138246@qq.com

**Data Availability Statement:** All relevant data are within the paper.

**Funding:** The author(s) received no specific funding for this work.

## Abstract

Depending on the trading modes, the effect of Outward Foreign Direct Investment (OFDI) on the manufacturing industry's position within the global value chain (GVC) may differ considerably. This paper examines the GVC position of China's manufacturing industry from 2003 to 2018, specifically focusing on the general trade and processing trade. Drawing upon this premise, this paper analyzes the effect and mechanism by which OFDI influences the GVC position of China's manufacturing industry. The result demonstrates that: (1) China's processing trade manufacturing industry has a much lower GVC position than general trade manufacturing industry. The GVC position of China's general trade manufacturing industry rose from 2.76 to 2.90 from 2003 to 2018, while processing trade manufacturing industry remained around 1.93. (2) OFDI boosts the GVC position of general trade manufacturing industry through facilitating reverse technology spillover, inducing industry structure upgrading, and enabling export scale expansion. (3) OFDI hinders the GVC position growth of processing trade manufacturing industry. The research findings offer theoretical backing for China to develop OFDI strategies that are tailored to different trading modes within the new framework of dual circulation. These strategies aim to facilitate the transformation and advancement of the manufacturing industry, as well as the growth of the GVC position.

## Introduction

The participation in global value chain (GVC) not only impacts a nation's capacity to derive advantages from global trade but also plays a crucial role in determining its economic transformation and advancement [1, 2]. China has actively integrated into GVCs through low-value-added, low-technology processing and production. However, this reliance on producing low- to mid-end products has a risk of becoming stuck in a cycle of low-quality production [3–5]. How to strengthen China's GVC position is crucial to advancing manufacturing industry transformation and upgrading under the dual circulation development pattern.

**Competing interests:** The authors have declared that no competing interests exist.

As a hub in the dual cycle, outward foreign direct investment (OFDI) links domestic and foreign markets. The expeditious growth of OFDI has proven advantageous for China's manufacturing industry in several ways. Firstly, it facilitates the integration of foreign capital, labor, and other resources, thereby enhancing the efficiency of domestic resource allocation [6]. Additionally, OFDI enables China's manufacturing industry to establish closer ties with foreign R&D [7]. In order to facilitate the acquisition of information pertaining to crucial production technologies and advanced management expertise, as well as to enhance its capacity for scientific and technology innovation and value creation [8]. China's OFDI topped that of the United States for the first time in 2020, ranking first globally. In 2021, China's OFDI reached $145.19 billion, a year-on-year increase of 9.2% [9]. Thus, China has become an important player in global OFDI.

It is noteworthy to mention that the manufacturing industry in China has a substantial size and is characterized by a dual trade structure [10, 11]. The dual trade structure primarily encompasses two dominant modes of trade, namely general trade and processing trade. The above two have differences in specific connotation. General trade refers to enterprises in China with import and export operation rights, engaging in unilateral import or unilateral export trade activities. Processing trade refers to the operations of enterprises that engage in the importation (or exportation) of raw and auxiliary materials, parts and components, components, packaging materials, and other materials. These materials are then processed or assembled, resulting in the production of finished goods that are then re-exported (or re-imported). According to China Customs figures, the average proportion of China's processing trade exports between 2002 and 2007 was recorded at 53.07%. Despite experiencing a gradual decline in recent years, this proportion remains as high as 20.1% in 2022. China's import and export commerce exhibits a distinct dual trade structure.

In the realm of manufacturing, notable distinctions exist between general trade and processing trade, particularly with regard to technological sophistication and profitability [12]. The general trade manufacturing industry exhibits a more intricate production chain, hence enhancing its productivity, capacity to absorb technology, and worldwide competitiveness. Conversely, the processing trade manufacturing industry is distinguished by the principles of "both ends out" and "large import, large export". The phrase "both ends out" refers to the scenario when the procurement of raw materials and the distribution of finished products occur in the global market. "Large import" refers to the raw materials required for the production of enterprises, mainly dependent on imports. "Large exports" means that enterprises' large-scale exports are based on large imports. It is evident that the aforementioned features pertaining to the processing trade manufacturing industry contribute to a diminished technological content in production and value added in trade [13–15].

Based on our analysis, it can be inferred that the differences between general trade manufacturing and processing trade manufacturing potentially contributes to a more pronounced divergence in the China's GVC position. This could also cause huge variances in OFDI's impact on China's manufacturing industry's GVC position under different trading forms. Therefore, understanding how OFDI affects the GVC position of general trade manufacturing industry and processing trade manufacturing industry would help China restructure and grow its manufacturing sector industry.

As economic globalization has advanced, more academics have concentrated on the effect of OFDI on the GVC. Based on their findings, pertinent studies can be split into two categories. On the one hand, some scholars believe that OFDI can enhance a country's GVC position both in the home and host countries [16–19]. For example, Yang & Luo found that OFDI has raised China's GVC position through reverse technology spillovers [20], While Liu et al. discovered that OFDI has helped China ascend in the GVC by promoting product quality and

functional upgrading [21]. Nie & Li indicated that the deindustrialization of the service sector has a significant positive effect on improving a firm's GVC position through OFDI [22]. The above related literature mainly reflects that OFDI has technology spillover effect, which further improves the GVC position. The theory of technology localization and the theory of technological innovation industrial upgrading provide theoretical support for the technology spillover effect of OFDI. In addition, the impact mechanism of OFDI on a country's GVC position also includes marginal industry transfer effects [18], export creation effects [23], market competition effects [24] and so on.

On the one hand, some scholars argue that OFDI may have neutral or negative effects on a country's GVC position [25]. This is mainly because OFDI may have substitution effects on domestic investment [26–28], leading to the crowding out of resources needed for domestic industry development, such as research and development investment. Additionally, OFDI aimed at marginal industry transfer may carry the risk of industrial hollowing-out [29, 30].

Unlike most countries, China has a relatively high proportion of processing trade in its foreign trade, which has gradually attracted the attention of the academic community. Processing trade enterprises generally have lower productivity and international competitiveness due to their low technological content and value-added features [31]. Therefore, China's processing trade and general trade exhibit significant differences in terms of trade transformation path [32], manufacturing service level [33], and production chain position. Wang et al. pointed out that processing trade is closer to downstream consumption in the production chain, and if general trade and processing trade are not distinguished in the measurement process, it may distort China's true GVC position [34]. To this end, Ma & Li and Peng & Wu further described the upstream index of China's manufacturing industry from the perspectives of general trade and processing trade. They found that China's GVC position of general trade was significantly higher than that of processing trade [35, 36].

Reviewing the relevant literature, it is found that most studies on the impact of OFDI on a country's GVC position view China's manufacturing industry as a whole. Few studies examining the impact of OFDI on China's manufacturing industry GVC position from the perspective of general trade and processing trade. Due to the significant differences between general trade and processing trade in terms of manufacturing efficiency, technology absorption ability, and international competitiveness, the impact of OFDI on the GVC position of general trade and processing trade manufacturing may differ, which has been overlooked in previous literature. Therefore, this paper uses national and regional input-output table data that differentiate between different trade modes to measure China's GVC position in general trade and processing trade manufacturing. Based on this, the paper examines the impact and mechanism of OFDI on the GVC position of manufacturing under different trade modes.

Compared with previous research, this paper has three innovations: First, based on the large proportion of processing trade in China, this paper uses the world input-output table data which distinguishes between general trade and processing trade, to calculate the GVC position of manufacturing under different trade modes. It will further refine the relevant research on China's GVC. Second, the large proportion of processing trade may have an impact on the general conclusion that OFDI promotes the position of GVC. Therefore, different from the perspective of previous research, this paper examines the impact of OFDI on the GVC position of the general trade manufacturing industry and the processing trade manufacturing industry based on the perspective of China's typical dualistic trade structure. Third, considering the differences between general trade and processing trade in terms of production efficiency and technology absorption capacity, this paper explores the mechanism of OFDI's impact on the GVC position of manufacturing under different trade modes from the aspects of reverse technology spillover, industry structure, and export scale.

## Theoretical analysis and research hypotheses

In order to assess the influence of OFDI on the GVC position of China's manufacturing industry, we undertakes a theoretical analysis through distinguishing between general trade and processing trade. In addition, the underlying mechanisms are elucidated by analyzing reverse technology spillovers, industry structure and export size.

## The impact of OFDI on the GVC position of the manufacturing industry under different trade modes

OFDI is a crucial element of the "dual circulation" strategy and links the domestic and international markets. The ability of China's manufacturing industry to integrate foreign technology, capital, knowledge, labor, and other resources, transfer surplus capacity, and optimize domestic resource allocation are all made possible by OFDI [37]. At the same time, through OFDI, China's manufacturing industry can enhance its trade relations with host countries, reduce trade costs, and further expand its foreign markets, thus climbing up the GVC. Due to technology, production, and other differences between general trade and processing trade, OFDI may affect the manufacturing industry's GVC position differently under different trading modes.

On the one hand, in China's general trade manufacturing industry, which engages in unilateral import or export activities, the exported products often satisfy the production needs of other countries, with complex production processes and high technological content. Through OFDI, the industry can effectively integrate technology, labor and other factors and optimize resource allocation, thus promoting the climbing up of the GVC. On the other hand, processing trade enterprises generally have lower production efficiency, technological content and technology absorption ability. Their OFDI behavior not only makes it more challenging to detect reverse technology spillover effects, but also increases the homogenous competition faced by parent companies, reducing their overseas market share, and hindering their climbing up of the GVC. Based on this, we propose Hypothesis 1.

Hypothesis 1: OFDI can significantly enhance the GVC position of the general trade manufacturing industry, but it has a restraining effect on the climbing up of the GVC for the processing trade manufacturing industry.

## The mechanism of the impact of different trade modes on the position of the manufacturing industry in the GVC through OFDI

**The reverse technology spillover effect of OFDI.** Under the new pattern of dual circulation, China's manufacturing industry relies on OFDI to embed itself in the international market cycle. With independent investment methods, regions, industries, and other channels, China integrates scarce factors and resources like technology and capital, improves resource allocation efficiency, promotes domestic circulation, and helps the manufacturing industry climb the GVC. Specifically, investment enterprises first achieve technology acquisition through two ways: one is cross-border mergers and acquisitions, which directly obtain advanced production technology and management experience from the acquired enterprises in the host country [38]; the other is to establish overseas subsidiaries or research and development institutions in the host country, integrate into the host country's domestic production link, and obtain the external effects of technology spillovers in the host country. In order to achieve the reverse technology spillover effect of OFDI, they first learn advanced technology from the host nation and then integrate it into the production of domestic products through imitation, demonstration, personnel flow, industry connection, and other channels [39].

**The industry structure upgrading effect of OFDI.** By selecting OFDI destination countries, China's manufacturing industry can transfer marginal industries in the internal circulation operating system that have advantages over the destination country to the host country to operate production links at low cost. The resulting capital, labor, and other production elements are reallocated. Domestic high-tech and rising manufacturing industries are developed, updating the country's industrial structure [40]. Moreover, through the expansion of production scale and the optimization of investment returns, Chinese manufacturing firms have the potential to reduce production costs and enhance the worldwide competitiveness of their products. Consequently, this can lead to an elevation in the level of their industrial structures and facilitate upward movement within the GVC. The upgrading of the manufacturing industry's industry structure in the subdivision industries is one way to convey the industrial structure upgrading effect.

**The expansion effect of the export scale of OFDI.** Under the new dual-circulation pattern, the Chinese manufacturing industry reduces production costs, transaction costs, and trade barrier costs by establishing subsidiaries or dividing the production process in host countries through OFDI. This increases China's commerce with host nations and lowers the export trade fixed cost ceiling, which helps China's manufactured exports grow in volume [41]. Numerous studies have also demonstrated that OFDI can significantly raise domestic productivity levels [42, 43]. It can help manufacturing companies to surpass the threshold of export trade costs and enhance their ability for autonomous innovation. Furthermore, it can stimulate the growth of export volume by offering more competitive products, thereby enhancing their GVC position. It is clear that employing OFDI, the Chinese manufacturing industry may reduce international trade costs and boost production effectiveness, thus increasing the export scale and climbing the GVC.

However, compared to general trade manufacturing, the production process of processing trade manufacturing mostly involves a large number of imported intermediate inputs, which are then exported after simple processing and assembly. The complexity of production technology is relatively low, and the ability to absorb technology is weak. This makes it difficult for processing trade enterprises to obtain foreign advanced technology and other resources through OFDI and conduct relevant industrial transfers. Because processing trade enterprises exports have a low value-added content, there is no room for OFDI to significantly lower manufacturing and trade expenses. In addition, China's processing trade manufacturing is in a low-end position in the GVC, and the processing and assembly of precision instruments are still controlled by developed countries. To maintain their interests and GVC leadership position, developed countries have set up various technical barriers, personnel mobility barriers, and market barriers for foreign investment, which are not conducive to the ascent of processing trade enterprises in the GVC through OFDI. Based on this, we proposes hypothesis 2.

Hypothesis 2: OFDI boosts general trade manufacturing's GVC position through reverse technology spillover, industrial structure upgrading, and export scale development, but not processing trade manufacturing.

## Measuring and analyzing the position of China's manufacturing industry in the GVC

Based on the world input-output model that distinguishes between general trade and processing trade, and with reference to Antràs output upstream degree index measurement method [44], the overall and trade-specific positions of China's manufacturing industry in the GVC are obtained. And then a dynamic analysis is conducted using a Gaussian kernel density function.

## Measurement model for the position of China's manufacturing industry in the GVC

According to the ICIO model, a world input-output table is constructed that distinguishes between general trade (G) and processing trade (P) activities in China (as shown in Table 1). It is assumed that there are (N+1) countries or regions in the world, and China is divided into two "regions" that engage in either general trade or processing trade. Each country or region has M industries, of which Q are manufacturing industries that produce specific products or services. It is worth noting that processing trade enterprises can only engage in production, processing, and export businesses, so processing trade cannot provide intermediate consumption goods for general trade, nor can it provide final consumption goods for China as a whole.

In Table 1, the superscripts G and P represent China's general trade and processing trade, respectively. Z is an (N+1) * (N+1) matrix representing the intermediate input and usage of goods among countries, Y and X are column vectors of (N+1) * 1, representing a country's final use (including household final consumption, non-profit institutions serving households, government final consumption, gross fixed capital formation, changes in inventories and valuables, and foreign purchases) and total output, respectively. VA is a row vector of 1 * (N+1), representing the value added that each country obtains in production. The superscript ´´´" indicates a transpose operation.

From the perspective of usage, when the market is cleared, the following equilibrium equation exists for Table 1:

$$\begin{bmatrix} 0 & 0 & Z^{P1} & \cdots & Z^{PN} \\ Z^{GP} & Z^{GG} & Z^{G1} & \cdots & Z^{GN} \\ Z^{1P} & Z^{1G} & Z^{11} & \cdots & Z^{1N} \\ \cdots & \cdots & \cdots & \ddots & \cdots \\ Z^{NP} & Z^{NG} & Z^{N1} & \cdots & Z^{NN} \end{bmatrix} + \begin{bmatrix} 0 & Y^{P1} & \cdots & Y^{PN} \\ Y^{GC} & Y^{G1} & \cdots & Y^{G} \\ Y^{1C} & Y^{11} & \cdots & Y^{1} \\ \cdots & \cdots & \ddots & \cdots \\ Y^{NC} & Y^{N1} & \cdots & Y^{NN} \end{bmatrix} = \begin{bmatrix} X^{P} \\ X^{G} \\ X^{1} \\ \cdots \\ X^{N} \end{bmatrix} \qquad (1)$$

Assuming the direct input coefficient $A^{NN} \equiv Z^{NN}(\hat{X}^{N})^{-1}$, where $(\hat{X}^{N})^{-1}$ is the inverse matrix of the diagonal matrix of N country's total output vector. Eq (1) can be expressed as:

$$\begin{bmatrix} 0 & 0 & A^{P1} & \cdots & A^{PN} \\ A^{GP} & A^{GG} & A^{G1} & \cdots & A^{GN} \\ A^{1P} & A^{1G} & A^{11} & \cdots & A^{1N} \\ \cdots & \cdots & \cdots & \ddots & \cdots \\ A^{NP} & A^{NG} & A^{N1} & \cdots & A^{NN} \end{bmatrix} \begin{bmatrix} X^{P} \\ X^{G} \\ X^{1} \\ \cdots \\ X^{N} \end{bmatrix} + \begin{bmatrix} 0 & Y^{P1} & \cdots & Y^{PN} \\ Y^{GC} & Y^{G1} & \cdots & Y^{G} \\ Y^{1C} & Y^{11} & \cdots & Y^{1} \\ \cdots & \cdots & \ddots & \cdots \\ Y^{NC} & Y^{N1} & \cdots & Y^{NN} \end{bmatrix} = \begin{bmatrix} X^{P} \\ X^{G} \\ X^{1} \\ \cdots \\ X^{N} \end{bmatrix} \qquad (2)$$

Let $A$, $X$, and $Y$ respectively represent the world direct input coefficient matrix, total output column vector, and final use column vector. Then Eq (2) can be written as:

$$X = AX + Y = (I - A)^{-1}Y \qquad (3)$$

In Eq (3), $(I-A)^{-1}$ is the Leontief inverse matrix. From Formula (3), the form of the infinite sequence of Formula (4) can be obtained:

$$X = Y + AY + AAY + \cdots\cdots \qquad (4)$$

The number of stages from the first item in Formula (4) to the final demand is 1, which is actually the number of production stages of the final product. The number of stages from the

**Table 1. World input-output table distinguishing China's general trade from its processing trade.**

| | | | Intermediate using | | | | | Final using | | | | Total out-put |
|---|---|---|---|---|---|---|---|---|---|---|---|---|
| | | | China (C) | | Country1 | ... | Country N | China (C) | Country1 | ... | Country N | |
| | | | Processing trade | General trade | | | | | | | | |
| Inter-mediate using | C | Process-ing trade | 0 | 0 | $Z^{P1}$ | ... | $Z^{PN}$ | 0 | $Y^{P1}$ | ... | $Y^{PN}$ | $X^P$ |
| | | General trade | $Z^{GP}$ | $Z^{GG}$ | $Z^{G1}$ | ... | $Z^{GN}$ | $Y^{GC}$ | $Y^{G1}$ | ... | $Y^{GN}$ | $X^G$ |
| | | Country 1 | $Z^{1P}$ | $Z^{1G}$ | $Z^{11}$ | ... | $Z^{1N}$ | $Y^{1C}$ | $Y^{11}$ | ... | $Y^{1N}$ | $X^1$ |
| | | ... | ... | ... | ... | ... | ... | ... | ... | ... | ... | ... |
| | | Country N | $Z^{NP}$ | $Z^{NG}$ | $Z^{N1}$ | ... | $Z^{NN}$ | $Y^{NC}$ | $Y^{N1}$ | ... | $Y^{NN}$ | $X^N$ |
| Value added | | | $VA^P$ | $VA^G$ | $VA^1$ | ... | $VA^N$ | — | — | — | — | — |
| Total input | | | $(X^P)'$ | $(X^G)'$ | $(X^1)'$ | ... | $(X^N)'$ | — | — | — | — | — |

second term to final demand is 2:1 intermediate goods production stage and 1 final goods production stage. Next, borrowing the research method of Alfaro & Chor, assuming that the distance between any two production stages in a country is equal and is 1 [45]. Then the output of the country $i$ industry $r$ can be expressed as:

$$x_r^i = y_r^i + \sum_{mj} a_{rn}^{ij} y_n^j + \sum_{mj}\sum_{sk} a_{rn}^{ij} a_{ns}^{jk} y_s^k + \cdots \tag{5}$$

In the above equation, $x$ represents total output, $a$ represents the coefficient of input demand, and $y$ represents total input. Continue to multiply the right-hand side of Eq (5) by the sum of the distances from their corresponding final consumption expenditures plus one and then divide by output. The weighted average position of a single industry in the production chain is thus calculated. The output upstream index of industry $r$ in country $i$ is obtained, as shown in Formula (6):

$$OUI_r^i = 1 \times \frac{y_r^i}{x_r^i} + 2 \times \frac{\sum_{mj} a_{rn}^{ij} y_n^j}{x_r^i} + 3 \times \frac{\sum_{mj}\sum_{sk} a_{rn}^{ij} a_{ns}^{jk} y_s^k}{x_r^i} + \cdots \tag{6}$$

Based on the upstream degree index of the $r$ industry output of $i$ country calculated in Eq (6), its position in the GVC can be determined. The larger the value $OUI$, the greater the proportion of intermediate product output supplied by the $r$ industry to other industries in its total output, and the more closely it is connected with other industries. In other words, the larger the value $OUI$, the closer the $r$ industry in $i$ country is to the production side of the production-consumption chain, which further makes its position in the GVC higher. Conversely, the smaller the value $OUI$, the closer the $r$ industry in $i$ country is to the final consumption side, and the lower its position in the GVC division of labor.

Furthermore, by simplifying Eq (6) and representing it in matrix form, we have:

$$OUI = \hat{X}^{-1}(Y + 2AY + 3A^2Y + \cdots) = \hat{X}^{-1}[(I - A)^{-1}]^2 Y \tag{7}$$

Based on Eq (6), multiplying the upstream degree index of the manufacturing $Q$ industries in $i$ country by the proportion of each industry in the total output of the manufacturing industry and then adding them up, we can obtain the upstream degree index of the overall manufacturing industry in $i$ country, as shown in Eq (8). It should be noted that this study assumes that China's general trade and processing trade are two "economic entities". Therefore, the upstream degree index of China's general trade and processing trade calculated using Eq (8) does not represent the GVC position of China's overall manufacturing industry. Therefore, when calculating the upstream degree index of China's overall manufacturing industry, the general trade and

processing trade of China in Table 1 should be merged first before calculating.

$$OUI^i = \sum_{r=1}^{m} OU_r^i \frac{x_r^i}{\sum_{r=1}^{m} x_r^i} \tag{8}$$

In addition, to clearly depict the dynamic evolution process of the GVC division of labor of China's overall manufacturing industry, general trade, and processing trade, Gaussian kernel density function is further used to analyze the position of the GVC division of labor curve, polarization trend, and extensibility changes. The Gaussian kernel density function expressions are shown in Eqs (9) and (10). $N$ is the sample size, $X_i$ and $x$ are the observed value and its mean value, respectively, $K(x)$ is the Gaussian kernel density function, and $h$ represents the bandwidth.

$$F(x) = \frac{1}{Nh} \sum_{i=1}^{N} K\left(\frac{X_i - x}{h}\right) \tag{9}$$

$$K(x) = \frac{1}{\sqrt{2\pi}} \exp\left(-\frac{x^2}{2}\right) \tag{10}$$

## Data source and processing

The most recent World Input-Output Tables (WIOT) for 2003–2018, which are compiled by the Organization for Economic Co-operation and Development (OECD) and distinguish China's trade modes, provide the data used to measure the overall, general trade, and processing trade GVC position of China's manufacturing industry. The WIOT adopts the ISIC Rev.4 classification standard, which divides each country's (region's) manufacturing industry into 17 sectors. As Chinese yearbook data uses the GB/T classification standard, we matches the ISIC Rev.4 industry with the GB/T industry according to China's National Economic Industry Classification released in 2017, which forms 16 sub-sectors of manufacturing industry. Following Wang classification criteria [34], we divides them into medium-low-tech and high-tech industries based on their technological intensity (see Table 2), to explore the impact of OFDI on China's manufacturing industry's integration into the GVC.

## Analysis of China's manufacturing industry's GVC position

As shown in Fig 1, from 2003 to 2018, China's manufacturing industry's GVC position showed a fluctuating upward trend, and the GVC position of processing trade manufacturing was much lower than that of general trade. In 2018, the upstream degree index of general trade was 2.90, which was 1.49 times that of processing trade. This is because processing manufacturing enterprises importing a large number of intermediate inputs from abroad, processing or assembling them, and exporting finished products with low value-added, technical content, and relevance to other industries. Additionally, the majority of products exported through processing trade are used to directly satisfy consumer demand, whereas products exported through general trade are mostly utilized to satisfy the production needs of other nations. The general trade manufacturing chain is longer and farther away from the consumer end than the processing trade chain. Therefore, it is important to make a distinction between general trade and processing trade when analyzing the GVC position of China's manufacturing industry.

Fig 2 depicts the dynamic development of the GVC position for the entire manufacturing sector in China as well as its sub-trade modes between 2003 and 2018. The peak of the distribution curve can be observed slowly migrating to the right in Fig 2(A) and 2(B), which shows that China's general trade manufacturing industry and overall manufacturing industry are

**Table 2. China's manufacturing industries are embedded in the GVC in different forms of trade.**

| Industry | | General trade | | | Processing trade | | |
|---|---|---|---|---|---|---|---|
| | | 2003 | 2010 | 2018 | 2003 | 2010 | 2018 |
| Medium-tech and low-tech industries | Food & Beverage & Tobacco manufacturing | 1.753 | 2.106 | 2.357 | 1.587 | 1.616 | 1.630 |
| | Textiles and clothing with leather and Related products manufacturing | 2.240 | 2.618 | 2.967 | 1.461 | 1.575 | 1.539 |
| | The lumber and softwood products industry | 2.947 | 3.237 | 3.155 | 2.695 | 2.693 | 2.640 |
| | Paper products and printing industry | 3.449 | 3.573 | 3.499 | 2.824 | 2.763 | 2.722 |
| | Furniture manufacturing and machinery and equipment repair and installation | 1.822 | 1.963 | 2.055 | 1.459 | 1.502 | 1.522 |
| | Coke and refined petroleum products industry | 3.713 | 3.842 | 3.772 | 2.577 | 2.726 | 2.676 |
| | Rubber and plastic products industry | 3.420 | 3.481 | 3.592 | 2.594 | 2.550 | 2.586 |
| | Non-metallic mineral products industry | 2.823 | 2.647 | 2.621 | 2.505 | 2.533 | 2.542 |
| | Basic metal products industry | 3.727 | 3.521 | 3.495 | 3.426 | 3.375 | 3.296 |
| | Metal products industry | 2.857 | 3.142 | 2.782 | 2.590 | 2.569 | 2.577 |
| High-tech industry | Chemical raw materials, fibers and chemical products industry | 3.613 | 3.848 | 4.011 | 2.910 | 2.999 | 2.953 |
| | Pharmaceutical products industry | 1.721 | 2.009 | 2.034 | 1.473 | 1.492 | 1.539 |
| | Machinery and equipment manufacturing | 2.129 | 2.262 | 2.340 | 1.821 | 1.876 | 1.857 |
| | Transportation equipment manufacturing industry | 2.632 | 2.084 | 2.155 | 1.712 | 1.561 | 1.561 |
| | Power equipment manufacturing industry | 2.624 | 2.292 | 2.712 | 2.192 | 2.035 | 2.122 |
| | Manufacturing of computer and electronic and optical products | 2.477 | 2.646 | 2.890 | 1.915 | 1.848 | 1.858 |

steadily moving up the GVC. In addition, from 2005 to 2015, a right peak appeared in the distribution curve, indicating that polarization occurred in the process of China's overall manufacturing industry and general trade manufacturing industry climbing up the GVC. However, starting in 2017, this differentiation phenomena started to lessen as seen by the right peak's steady slowing and the separation from the main peak reducing. The GVC distribution curve for China's processing trade manufacturing industry is depicted in Fig 2(C). There were two main peaks on the curve and the main peak of the curve did not exhibit any obvious movement, indicating a significant polarization phenomenon. And it means that processing trade manufacturing industries have significant disparities in their GVC positions.

The position of China's manufacturing industry in various trade forms integrated into the GVC is shown in Table 2. It is clear that from 2003 to 2018, 12 industries, such as those producing food, beverages, and tobacco, textile, apparel, and leather products, pharmaceuticals, and computer, electronic, and optical products, have significantly improved their GVC position in the form of general trade. While the GVC position of metal products, basic metal products, and non-metallic mineral goods has experienced a fall. Additionally, the position of

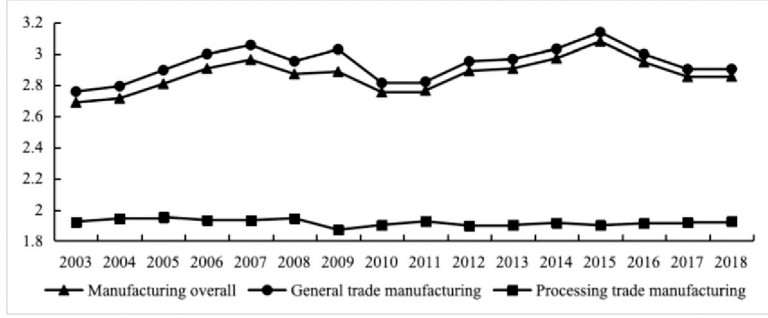

**Fig 1. GVC position of China's manufacturing industry in different trade modes.** Data source: drawn based on the calculated upstream index of China's manufacturing GVC.

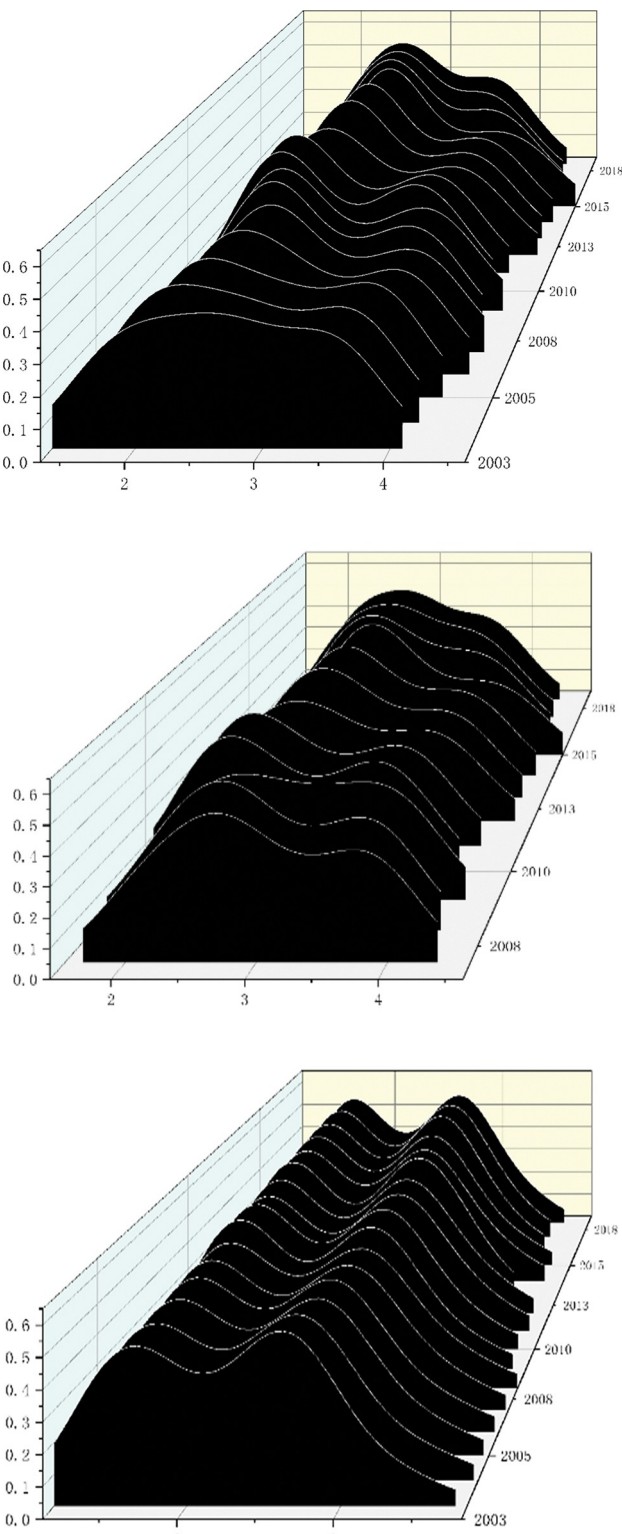

**Fig 2. Dynamic evolution of GVC position of different trade modes in manufacturing industry.** Data source: drawn based on the calculated upstream index of China's manufacturing GVC. (a) The overall GVC position distribution dynamics of Chinese manufacturing industry. (b) The GVC position distribution dynamics of general trade manufacturing industry. (c) The GVC position distribution dynamics of processing trade manufacturing industry.

different industries in the GVC in terms of processing trade has decreased significantly from 2003 to 2018, with half of the industries having done so significantly. Industries like those that manufacture transportation equipment, electric power equipment, and computer, electronic, and optical products are among those that have seen a decrease in processing trade position in the GVC of more than 3%.

## Model specification, variable selection, and data description

A literature review revealed that there is no research on OFDI's impact on China's manufacturing industry's GVC position from general trade and processing trade perspectives. To close this gap, this study builds a baseline regression model and a mechanism analysis model to investigate the impact and mechanism of OFDI on the GVC position and trade mode of China's manufacturing industry.

### Model construction

In order to investigate the effects of OFDI on the GVC position of China's manufacturing industry, this study builds a baseline regression model, as illustrated in Eq (11):

$$LNGVC\_pos_{i,t} = \beta_0 + \beta_1 OFDI_{i,t} + \beta_2 X_{i,t} + \alpha_i + \alpha_t + \mu_{i,t} \tag{11}$$

*i* and *t* in the preceding equation denote manufacturing industry sub-sectors and years, respectively. The industry and time control effects are represented by $a_i$ and $a_t$, respectively, while the disturbance term is represented by $\mu_{i,t}$. *GVC_pos* symbolizes the manufacturing industry's GVC position, whereas *OFDI* denotes the amount of Chinese OFDI in the manufacturing industry. *X* indicates the variables under control.

To further investigate the mechanism of OFDI on China's integration into the GVC, this study constructs a mechanism analysis model that includes interaction terms between OFDI and measures of reverse technology spillover effects, industry structure upgrading effects, and export scale effects, as shown in Eqs (12), (13) and (14):

$$LNGVC\_pos_{i,t} = \beta_0 + \beta_1 OFDI_{i,t} + \beta_2 LNPAT_{i,t} + \beta_3 OFDI_{i,t} \times LNPAT_{i,t} + \beta_4 X_{i,t} + \alpha_i + \alpha_t$$
$$+ \mu_{i,t} \tag{12}$$

$$LNGVC\_pos_{i,t} = \beta_0 + \beta_1 OFDI_{i,t} + \beta_2 LNRVA_{i,t} + \beta_3 OFDI_{i,t} \times LNRVA_{i,t} + \beta_4 X_{i,t} + \alpha_i + \alpha_t$$
$$+ \mu_{i,t} \tag{13}$$

$$LNGVC\_pos_{i,t} = \beta_0 + \beta_1 OFDI_{i,t} + \beta_2 EX_{i,t} + \beta_3 OFDI_{i,t} \times EX_{i,t} + \beta_4 X_{i,t} + \alpha_i + \alpha_t + \mu_{i,t} \tag{14}$$

*LNPAT*, *LNRVA* and *EX* in the above equations reflect reverse technology spillover effects, industry structure upgrading effects, and export scale expansion effects, respectively. The remaining variables have the same meaning as before.

### Variable selection

**Dependent variables.** *GVC_pos* represents the GVC position of the manufacturing industry, including *GVC_tot* (the overall GVC position of China's manufacturing industry), *GVC_gen* (the GVC position of manufacturing industries under general trade), and *GVC_pro* (the GVC position of manufacturing industries under processing trade). They are measured using the upstreamness index of manufacturing industry sub-sectors, as mentioned earlier.

**Key independent variables.** *OFDI* indicates the level of China's OFDI in the manufacturing industry and is calculated by dividing the corresponding year's OFDI flow by the logarithm. This variable directly reflects the changes in China's OFDI in the manufacturing industry and accurately analyzes its impact on the GVC position of the manufacturing industry.

**Mechanism analysis variables.** Reverse technology spillover effects (*LNPAT*). The frequency of patent applications for inventions in a country can represent its ability for innovation and, to some extent, quantify the reverse technology spillover benefits obtained through OFDI. To assess reverse technology spillover effects, the logarithm of the number of patent applications for inventions in each manufacturing industry sub-sector is employed. Industry structure upgrading effects (*LNRVA*). The logarithm of the proportion of value contributed in each manufacturing industry sub-sector to total value added was used as the measure for analyzing the industry structure upgrading effects of OFDI in this study. Export scale expansion effects (*EX*). To determine how OFDI affects export size in each manufacturing industry sub-sector, the proportion of export delivery value to total delivery value is employed.

**Control variables.** The control variable X includes the following variables: (1) research and development investment (*RD*), expressed as a proxy variable for R&D investment by the proportion of R&D expenditures in total output value of various sub-sectors of the manufacturing industry; (2) human capital (*HC*), defined as the ratio of total scientific and technological personnel to total employees in various sub-sectors of the manufacturing industry; (3) capital intensity (*CAP*), represented by the per capita capital stock for each sub-sector of the manufacturing industry; (4) industry size (*SCA*), measured by the ratio of each sub-sector's output value to the total output value of the manufacturing industry in the same year; (5) fixed asset investment (*FAI*), expressed by the logarithm of the total fixed assets of each manufacturing industry subsector.

**Data sources and processing.** This study examines the period from 2003 to 2018. In each yearbook, the matching and merging of sub-sectors in the manufacturing industry is done as detailed in the previous section. The stock data for OFDI in the manufacturing industry is derived from the corresponding year's "Statistical Bulletin of Outward FDI". R&D expenditures, patent applications, total number of scientific and technology workers, total number of employees, and other data are taken from the "China Science and Technology Statistical Yearbook" for each year. Furthermore, according to Yu's proposed depreciation rate calculation technique, the capital stock of each sub-sector in the manufacturing industry is computed using the perpetual inventory method, with data from the "China Statistical Yearbook" being used [46]. The "China Industrial Statistical Yearbook" provides data on export delivery value for each sub-sector of the manufacturing industry. Table 3 displays descriptive statistics for each variable.

## Empirical results analysis

To investigate the impact of OFDI on the GVC position of the manufacturing industry under different trade modes, the previous section's regression model is examined, and the mechanism is tested from three perspectives: reverse technology spillover effect, industry structure upgrading effect, and export scale expansion effect.

### Benchmark regression

The regression findings of OFDI on the GVC position of China's total manufacturing industry, general trade manufacturing industry, and processing trade manufacturing sector are shown in columns (1), (3), and (5) of Table 4. Columns (2), (4), and (6) of Table 4 indicate the

**Table 3. Descriptive statistics of each variable.**

| Variable | meaning | Sample size | mean | standard deviation | minimum | maximum |
|---|---|---|---|---|---|---|
| GVC_tot | Manufacturing GVC position | 256 | 2.886 | 0.678 | 1.718 | 4.272 |
| GVC_gen | General trade manufacturing GVC position | 256 | 2.935 | 0.662 | 1.721 | 4.283 |
| GVC_pro | Processing trade manufacturing GVC position | 256 | 2.234 | 0.595 | 1.459 | 3.493 |
| OFDI | Outward foreign direct investment | 256 | 0.787 | 1.001 | -0.714 | 2.402 |
| LNPAT | Foreign direct investment | 256 | 7.816 | 1.762 | 1.946 | 11.599 |
| LNRVA | Manufacturing industry structure | 256 | -3.982 | 1.506 | -6.763 | -0.569 |
| EX | Scale of export | 256 | 0.063 | 0.093 | 0.004 | 0.459 |
| RD | R&d investment | 256 | 0.009 | 0.006 | 0.001 | 0.025 |
| HC | Human capital | 256 | 0.035 | 0.021 | 0.004 | 0.099 |
| CAP | Degree of capital intensity | 256 | 10.862 | 10.216 | 0.000 | 66.259 |
| SCA | Size of industry | 256 | 0.063 | 0.033 | 0.013 | 0.123 |
| FAI | Investment in fixed assets | 256 | 8.819 | 0.909 | 6.012 | 10.592 |

regression findings in the previous regression with the addition of control variables. Columns (1) and (2) show that OFDI may considerably boost China's manufacturing industry's GVC position. Furthermore, R&D investment and industry size have a major beneficial effect on China's manufacturing industry's ascension in the GVC, whereas fixed asset investment has a suppressive effect. This could be owing to technical restrictions imposed by industrialized countries on China, as well as the rapid expansion of indigenous emergent sectors, which have consumed a considerable amount of money.

**Table 4. Benchmark regression.**

| Variable | GVC_tot | | GVC_gen | | GVC_pro | |
|---|---|---|---|---|---|---|
| | (1) | (2) | (3) | (4) | (5) | (6) |
| OFDI | 0.061*** | 0.204*** | 0.059*** | 0.209*** | -0.003 | -0.046*** |
| | (0.018) | (0.045) | (0.019) | (0.049) | (0.007) | (0.016) |
| RD | | 31.854*** | | 25.722*** | | 10.813*** |
| | | (6.063) | | (6.568) | | (2.223) |
| HC | | 1.056 | | 0.389 | | 0.049 |
| | | (1.106) | | (1.209) | | (0.403) |
| CAP | | 0.000 | | 0.000 | | 0.002** |
| | | (0.002) | | (0.002) | | (0.001) |
| SCA | | 4.057** | | 4.991*** | | -0.771*** |
| | | (1.690) | | (1.746) | | (0.277) |
| FAI | | -0.345*** | | -0.333*** | | 0.053* |
| | | (0.078) | | (0.083) | | (0.028) |
| Cons | 2.751*** | 4.939*** | 2.789*** | 4.905*** | 2.232*** | 1.719*** |
| | (0.026) | (0.604) | (0.028) | (0.654) | (0.009) | (0.223) |
| Industry fixed effects | Yes | Yes | Yes | Yes | Yes | Yes |
| Year fixed effect | Yes | Yes | Yes | Yes | Yes | Yes |
| R² | 0.349 | 0.473 | 0.347 | 0.443 | 0.138 | 0.282 |
| N | 256 | 256 | 256 | 256 | 256 | 256 |

Note: The values in brackets in the table are standard errors.

***, ** and * indicate significance at the confidence levels of 1%, 5% and 10%. Same as below.

Columns (4) and (5) of Table 4 show that, when different trade modes are considered, OFDI can increase the GVC position of the general trade manufacturing sector while inhibiting the ascent of the processing trade manufacturing business. It can be shown that OFDI boosts China's manufacturing sector's GVC position by promoting general trade manufacturing.The first hypothesis has been proven.

## Dealing with endogeneity issues

Due to the reverse causation impact between a country's manufacturing GVC and OFDI, the baseline model may have endogeneity issues [47]. Furthermore, despite the fact that the regression model contains numerous control variables, there is still the risk of omitted variable bias, which may alter the estimation findings. To address endogeneity difficulties, this study employs the System GMM and 2SLS approaches in the instrumental variable approach to assure reliable model estimate. In terms of instrumental variable selection, this study uses Zhang & Yu [48] lagged first-order of the core explanatory variable. Table 5 displays the regression results.

Table 5 shows that, using both the System GMM and 2SLS methodologies, OFDI increases China's manufacturing industry's overall position and the GVC position of general trade manufacturing, but suppresses the ascent of processing trade manufacturing in the GVC. This is consistent with the result of the baseline regression.

**Table 5. System GMM and 2SLS test.**

| Variable | GMM | | | 2SLS | | |
|---|---|---|---|---|---|---|
| | (1) | (2) | (3) | (4) | (5) | (6) |
| | GVC_tot | GVC_gen | GVC_pro | GVC_tot | GVC_gen | GVC_pro |
| L.GVC_tot | 0.883*** | | | | | |
| | (0.042) | | | | | |
| L.GVC_gen | | 0.643*** | | | | |
| | | (0.060) | | | | |
| L.GVC_pro | | | 0.302*** | | | |
| | | | (0.100) | | | |
| OFDI | 0.061*** | 0.064** | -0.063** | 0.281*** | 0.251** | -0.014 |
| | (0.017) | (0.025) | (0.025) | (0.092) | (0.101) | (0.034) |
| RD | -18.292*** | -28.979*** | 3.222 | 29.666*** | 23.874*** | 10.807*** |
| | (3.943) | (5.648) | (2.299) | (6.003) | (6.584) | (2.223) |
| HC | 2.481*** | 2.581*** | 0.452 | 1.334 | 0.610 | 0.213 |
| | (0.646) | (0.961) | (0.422) | (1.060) | (1.177) | (0.392) |
| CAP | -0.001 | 0.001 | -0.002* | 0.000 | 0.001 | 0.001* |
| | (0.002) | (0.003) | (0.001) | (0.002) | (0.002) | (0.001) |
| SCA | 2.189** | 2.424* | 0.134 | 3.243** | 4.184** | -0.653** |
| | (0.935) | (1.436) | (0.494) | (1.650) | (1.711) | (0.296) |
| FAI | -0.069** | 0.053 | 0.105** | -0.306*** | -0.303*** | 0.059** |
| | (0.033) | (0.050) | (0.044) | (0.077) | (0.083) | (0.028) |
| Cons | 0.921*** | 0.671** | -0.312 | | | |
| | (0.197) | (0.293) | (0.432) | | | |
| Industry fixed effects | Yes | Yes | Yes | Yes | Yes | Yes |
| Year fixed effect | Yes | Yes | Yes | Yes | Yes | Yes |
| Sargan test | 0.464 | 0.492 | 0.526 | | | |
| Weak instrumental variable test | | | | 27.100 | 27.100 | 21.086 |
| N | 256 | 256 | 256 | 240 | 240 | 240 |

**Table 6. Robustness test.**

| Variable | Replace the core explanatory variables | | | FGLS | | |
|---|---|---|---|---|---|---|
| | (1) | (2) | (3) | (4) | (5) | (6) |
| | GVC_tot | GVC_gen | GVC_pro | GVC_tot | GVC_gen | GVC_pro |
| OFDI | 0.145*** | 0.149*** | -0.033*** | 0.109*** | 0.111*** | -0.014 |
| | (0.032) | (0.035) | (0.012) | (0.041) | (0.041) | (0.012) |
| RD | 31.854*** | 25.722*** | 10.813*** | 12.167*** | 5.632 | 2.418 |
| | (6.063) | (6.568) | (2.223) | (4.146) | (4.509) | (1.545) |
| HC | 1.056 | 0.389 | 0.049 | 0.700 | 0.147 | -0.109 |
| | (1.106) | (1.209) | (0.403) | (0.772) | (0.779) | (0.239) |
| CAP | 0.000 | 0.000 | 0.002** | 0.002 | 0.002 | 0.001 |
| | (0.002) | (0.002) | (0.001) | (0.002) | (0.002) | (0.001) |
| SCA | 4.057** | 4.991*** | -0.771*** | 3.133** | 3.882*** | -0.516** |
| | (1.690) | (1.746) | (0.277) | (1.395) | (1.394) | (0.258) |
| FAI | -0.345*** | -0.333*** | 0.053* | -0.147** | -0.117 | 0.019 |
| | (0.078) | (0.083) | (0.028) | (0.072) | (0.073) | (0.021) |
| Cons | 4.723*** | 4.684*** | 1.769*** | 2.945*** | 2.602*** | 1.431*** |
| | (0.562) | (0.609) | (0.208) | (0.608) | (0.628) | (0.176) |
| Industry fixed effects | Yes | Yes | Yes | Yes | Yes | Yes |
| Year fixed effect | Yes | Yes | Yes | Yes | Yes | Yes |
| N | 256 | 256 | 256 | 256 | 256 | 256 |

## Robustness checks

To validate the empirical conclusions, this study replaces the baseline regression's primary explanatory variable with China's OFDI stock's natural logarithm in the following year. Furthermore, to avoid heteroscedasticity and intergroup correlation in panel data, robustness assessments use the generalised least squares (FGLS) estimation method. Table 6 shows findings.

Columns (1) to (3) show the regression results with the main explanatory variable replaced, while columns (4) to (6) show the results with the estimate technique replaced. We can see that OFDI can significantly improve China's manufacturing industry's overall position and its position in the GVC embedded in general trade, but it suppresses processing trade manufacturing's rise in the GVC. This suggests that the empirical findings are sound and reliable.

## Heterogeneity analysis

To examine the impact of OFDI on the GVC rank of manufacturing industries of various technological categories, binary dummy variables with values of 1 for high-tech industries and 0 for medium-to-low-tech sectors are created. Table 7 shows the results of incorporating the interaction term into the regression model. The coefficient of the interaction term in columns (1) and (2) is considerably positive, showing that OFDI promotes the GVC position of high-tech manufacturing businesses more than medium-to-low-tech industries. Column (3) reveals that the interaction term coefficient is not significant, indicating that OFDI suppresses high-tech and medium-to-low-tech manufacturing enterprises in processing trade similarly.

## Mechanism analysis

As seen by the previous regression results, the influence of OFDI on the manufacturing industry's GVC position varies dramatically between trade modes. A mechanism analysis is

**Table 7. Heterogeneity test.**

| Variable | (1) | (2) | (3) |
|---|---|---|---|
| | GVC_tot | GVC_gen | GVC_pro |
| OFDI | 0.216*** | 0.214*** | -0.046*** |
| | (0.044) | (0.048) | (0.010) |
| OFDI×TEC | 0.058*** | 0.049** | 0.011 |
| | (0.018) | (0.019) | (0.007) |
| RD | 39.696*** | 31.941*** | 9.462*** |
| | (6.392) | (6.938) | (2.077) |
| HC | -0.085 | -0.577 | -0.249 |
| | (1.136) | (1.254) | (0.403) |
| CAP | 0.000 | 0.000 | 0.001* |
| | (0.002) | (0.002) | (0.001) |
| SCA | 4.910*** | 5.065*** | -0.813*** |
| | (1.673) | (1.725) | (0.300) |
| FAI | -0.400*** | -0.368*** | -0.001 |
| | (0.078) | (0.083) | (0.022) |
| Cons | 5.291*** | 5.157*** | 2.253*** |
| | (0.601) | (0.654) | (0.192) |
| Industry fixed effects | Yes | Yes | Yes |
| Year fixed effect | Yes | Yes | Yes |
| $R^2$ | 0.498 | 0.458 | 0.230 |
| N | 256 | 256 | 256 |

performed from the perspectives of reverse technology spillovers, industry structure, and export scale to further investigate the reasons for these disparities. The mechanism analysis is carried out in two stages. The basic explanatory variables are regressed against patent applications, industry structure indicators, and export scale in the first stage. The results reveal that OFDI can produce reverse technology spillover effects, industry structure upgrading effects, and export scale expansion effects. This conclusion is similar with the findings of Yang & Luo, Yue and Zhang et al. [7, 8, 20]. The second stage is to regress the mechanism variables' OFDI and interaction terms against the manufacturing industry's GVC position. Table 8 displays the results.

The findings of assessing the reverse technology spillover effects, industry structure upgrading effects, and export scale expansion impacts of OFDI on the GVC position of the manufacturing industry are shown in columns (1)-(3), (4)-(6), and (7)-(9) of Table 8. Except for the negligible interaction factors in columns (3), (6), and (9), it can be seen that all other interaction variables are significant. This suggests that while OFDI can improve the overall and general trade manufacturing GVC position through reverse technology spillover effects, industry structure upgrading effects, and export scale expansion effects, its effect on processing trade manufacturing is negligible, limiting its upward mobility in the GVC. Hypothesis 2 is confirmed. This conclusion not only highlights the difference between general trade and processing trade in terms of technical content [11, 15, 31], indirectly supports previous research on the promotion of a country's or region's GVC position by OFDI [19, 49, 50], but also broadens the research boundary of OFDI and GVC, allowing for further refinement of the research.

Based on this, OFDI may suppress the upward mobility of processing trade manufacturing in the GVC by reducing the home enterprises' market share in overseas markets and exacerbate

**Table 8. Mechanism test.**

| Variable | Reverse technology spillover effect | | | Industrial structure upgrading effect | | | Export scale expansion effect | | |
|---|---|---|---|---|---|---|---|---|---|
| | (1) | (2) | (3) | (4) | (5) | (6) | (7) | (8) | (9) |
| | GVC_tot | GVC_gen | GVC_pro | GVC_tot | GVC_gen | GVC_pro | GVC_tot | GVC_gen | GVC_pro |
| OFDI×LNPAT | 0.023*** | 0.019** | 0.003 | | | | | | |
| | (0.007) | (0.007) | (0.003) | | | | | | |
| OFDI×LNRVA | | | | 0.019*** | 0.013** | 0.002 | | | |
| | | | | (0.006) | (0.006) | (0.002) | | | |
| OFDI×EX | | | | | | | 0.465*** | 0.362*** | 0.047 |
| | | | | | | | (0.089) | (0.099) | (0.034) |
| Cons | 5.429*** | 5.291*** | 1.833*** | 4.899*** | 4.829*** | 1.717*** | 4.264*** | 4.415*** | 2.127*** |
| | (0.608) | (0.660) | (0.221) | (0.593) | (0.655) | (0.224) | (0.604) | (0.678) | (0.204) |
| Core explanatory variable | Yes | Yes | Yes | Yes | Yes | Yes | Yes | Yes | Yes |
| Mechanism variable | Yes | Yes | Yes | Yes | Yes | Yes | Yes | Yes | Yes |
| Control variable | Yes | Yes | Yes | Yes | Yes | Yes | Yes | Yes | Yes |
| Industry fixed effects | Yes | Yes | Yes | Yes | Yes | Yes | Yes | Yes | Yes |
| Year fixed effect | Yes | Yes | Yes | Yes | Yes | Yes | Yes | Yes | Yes |
| $R^2$ | 0.505 | 0.464 | 0.317 | 0.508 | 0.456 | 0.285 | 0.534 | 0.475 | 0.234 |
| N | 256 | 256 | 256 | 256 | 256 | 256 | 256 | 256 | 256 |

homogenised competition, which hinders resource allocation like R&D costs [51, 52] and the enterprise's GVC position.

## Conclusion and Implications

### Conclusion

This paper examines the GVC position of China's manufacturing industry from 2003 to 2018, specifically focusing on the general trade and processing trade. Drawing upon this premise, this paper analyzes the effect and mechanism by which OFDI influences the GVC position of China's manufacturing industry. The result shows that: (1) China's processing trade manufacturing industry has a much lower GVC position than general trade manufacturing industry. The GVC position of China's general trade manufacturing industry rose from 2.76 to 2.90 from 2003 to 2018, while processing trade manufacturing industry remained around 1.93. (2) OFDI can boost general trade manufacturing's GVC position while hindering processing trade manufacturing's rise. This conclusion remains after a variety of robustness tests and consideration of endogeneity issues. (3) Mechanism analysis shows that OFDI promotes the GVC position of the general trade manufacturing industry through reverse technology spillover, industrial structure upgrading, and export scale expansion, but not the processing trade manufacturing industry. It may also influence conclusion (2).

According to the research findings, the promotion effect of OFDI on the GVC position of manufacturing industry is primarily concentrated in the general trade manufacturing industry, rather than the processing trade manufacturing business. As a result, when a country, such as China, has a high proportion of processing trade, it must evaluate the influence of diverse trade modes while evaluating issues related to its GVC. Similarly, when analyzing the impact of OFDI on global value chain status, we need differentiate between regular commerce and processing trade. Our research refines relevant research on OFDI and the GVC, and provides a reference for China to create OFDI strategy under the new pattern of twofold circulation, in order to climb the GVC and realize manufacturing industry transformation and upgrading.

## Implications

First, China should develop OFDI in general trade manufacturing industry. China can promote the GVC position of manufacturing industry by taking advantage of the "Belt and Road" initiative, supporting domestic general trade manufacturing enterprises to perform OFDI, and developing differentiated investment strategies for different host countries.

Second, China should increase the scale of investment in high-tech businesses while facilitating dual circulation in local and international markets. On the one hand, build and improve special funds and policies for the "going global" plan, encouraging OFDI in high-tech sectors. On the other hand, organize regular bank-enterprise matchmaking sessions to solve their financing challenges.

Third, China should strengthen technology absorption and innovation capabilities. On the one hand, raise internal R&D expenditures while also establishing and improving a talent mobility system and training mechanism. On the other hand, adopt policies such as investment subsidies and credit support, to encourage the processing trade manufacturing industry to participate in the GVC, and promote its transformation and upgrading.

One of the important limitations of the study is that we could not find the reason why OFDI inhibits China's processing trade manufacturing industry from climbing the global value chain. we left this question for future research.

## Author Contributions

**Data curation:** Dong Le.

**Formal analysis:** Ziyu Hu.

**Writing – original draft:** Fei Ren.

**Writing – review & editing:** Ziyu Hu.

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
