## [Decision Letter · Decision Letter 0]

24 Jul 2023

PONE-D-23-19895Outward Foreign Direct Investment and GVC Position of Manufacturing Industry: A Perspective on China's Typical Binary Trade StructurePLOS ONE

Dear Dr. Hu,

Thank you for submitting your manuscript to PLOS ONE. After careful consideration, we feel that it has merit but does not fully meet PLOS ONE’s publication criteria as it currently stands. Therefore, we invite you to submit a revised version of the manuscript that addresses the points raised during the review process.

As you can see the reviewers have now given detailed comments and recommended substantial modification and improvement in the the manuscript. There are several issues that need to be addressed before possible acceptance of the paper. Kindly revise your manuscript according to raised concerns and resubmit the paper again. 

We look forward to receiving your revised manuscript.

Kind regards,

Shujahat Haider Hashmi, PhD Regional Economics

Academic Editor

PLOS ONE

2. Please amend your authorship list in your manuscript file to include authors Ziyu Hu, Fei Ren and Dong Le.

Additional Editor Comments:

As you can see the reviewers have now given detailed comments and recommended major modification and improvement in the the manuscript. There are several issues that need to be addressed before possible acceptance of the paper. Kindly revise your manuscript according to raised concerns and resubmit the paper again.

Reviewers' comments:

Reviewer's Responses to Questions

**Comments to the Author**

1. Is the manuscript technically sound, and do the data support the conclusions?

Reviewer #1: Yes

Reviewer #2: Yes

2. Has the statistical analysis been performed appropriately and rigorously? 

Reviewer #1: Yes

Reviewer #2: Yes

3. Have the authors made all data underlying the findings in their manuscript fully available?

Reviewer #1: Yes

Reviewer #2: Yes

4. Is the manuscript presented in an intelligible fashion and written in standard English?

Reviewer #1: No

Reviewer #2: Yes

5. Review Comments to the Author

Reviewer #1: This study examines the impact and mechanism of outward FDI on the GVC position of China's manufacturing industry from the perspective of a typical binary trade structure. The research topic is very important and meaningful. However, there are still some weaknesses for authors before consulting publishing.

(1) The abstract can be further refined. For example, the research results in the abstract can be summarized into three points.

 2）The title of Figure 1 is too long, you should make it shorter, also for Figure 2.

(3) It is suggested to further improve the innovation points.

(4) Why you divide the industries to 2 kinds? Medium and low tech industries, High tech industry, why not 3 kinds?

(5) Some fresh papers can be added into references, eg:

Liu, F., Sim, J. Y., Sun, H., Edziah, B. K., Adom, P. K., & Song, S. (2023). Assessing the role of economic globalization on energy efficiency: Evidence from a global perspective. China Economic Review, 77, 101897. https://doi.org/10.1016/j.chieco.2022.101897.

Reviewer #2: PONE-D-23-19895

Outward Foreign Direct Investment and GVC Position of Manufacturing Industry: A Perspective on China's Typical Binary Trade Structure

1) Thank you for your interest to contribute into the PLOS journal

2) Main idea and topic

The research idea is interesting indeed. However, the title needs to showcase the elements of general trade and processing trade. Thus, the title need improvements.

3) Abstract

There is a lack on explaining the results which only shows that the GVC position of China's processing trade manufacturing 11 industry is far lower than that of the general trade manufacturing industry. A more specific is needed to highlight any figures or trends that revealed such results.

4) Introduction and literature review

a) In this paper, it has a strong background and clear direction. Furthermore its objectives are justified clearly.

b) However, some important elements that have been highlighted in the paper were not been stressed in the

introduction such as the elements of reverse technology spillover, structure upgrading and export scale

expansion.

c) Why the paper used the binary trade structure?

d) Elaborate further the world input-output model. How this model is in line to extract that results in this study?

5) Research methodology / results / Data source / analysis

a) Author did well in explaining the analysis and measure to extract the data particularly the general trade and

processing trade GVC.

b) In this paper, it has a strong background and clear direction. Furthermore its objectives are justified clearly.

c) The author needs source on a statement highlighted about "China Science and Technology Statistical Yearbook".

d) As for the mechanism analysis, author needs to add evidences of similarities or differences from this analysis

against previous studies.

e) Interestingly, processing manufacturing seems to not change significantly. Explain.

6) Discussion and conclusion

Conclusion is found to weak and has less aspect of generalization on the impact and contribution of the results in particular to China and the industry. Further a more comprehensive view of the study are needed to conquer the overall picture.

Conclusion needs to add the overall results and unique findings that can contribute to the knowledge area.

6. PLOS authors have the option to publish the peer review history of their article (what does this mean?). If published, this will include your full peer review and any attached files.

Reviewer #1: No

Reviewer #2: **Yes: **Mohd Haniff Jedin

---

## [Author Response · Author response to Decision Letter 0]

3 Sep 2023

Response to the Editor and Referee

Title: Outward Foreign Direct Investment and GVC Position of Manufacturing Industry: A Perspective on China's Typical Binary Trade Structure

Authors: Fei Ren, Dong Le, Ziyu Hu* 

Dear Prof :

We really appreciate your response concerning the initial submission of this paper to PLOS ONE. We also would like to express great appreciation to you and the referee for these constructive comments. We have thoughtfully taken into these comments. The explanations of what we have changed in response to the referee’ comments are given point by point. We hope that all these changes fulfill your requirements as well as the referee. Correspondence and calls about this paper should be directed to author at the following address and e-mail. Thanks very much again for your attention to our paper.

Once again, thank you for your time and help to our paper processing.

Yours Sincerely,

Ziyu Hu

Address: Economics School, Zhongnan University of Economics and Law, Wuhan 430073, China;

E-mail:994138246@qq.com

For your guidance, itemized responses to referee’s comments are appended below. 

1. Overall

Firstly, we are thankful to the referee for the highly instructive comments and valuable suggestions, which have been considered in this revision. Secondly, we correct our typing errors, grammatical errors in the earlier version of our manuscript. Thirdly, we modified the formats according to published papers to comply with journal requirements. The modifications that we have made to the manuscript are summarized as follows:

(1) All of the comments of the referee have been positively taken into account. 

(2) The problems of expression and grammatical errors have been addressed carefully.

(3) The formats have been modified according to published papers.

(4) In order to be more precise and convincible throughout the text, we have reexamined the full text and have revised the Abstract, Introduction, Theoretical support and model, Results, Discussion, Conclusion and Policy Implications. More latest and instructive literatures are cited in our revised manuscript. In addition, based on the empirical results, we have re-summarized the conclusions, impacts and contributions of the paper.

2. Detailed responses 

Thank you very much for reviewing the revised manuscript. We believe that your further comments and suggestions are highly constructive and very helpful for our research. We have thoughtfully taken into these comments and responded to your constructive suggestions from point by point outlined below. We hope we have addressed all of your concerns.

For the sake of presentation, the comments of the referee are reproduced in italics, and our responses are given in plain. 

Comments from the reviewers

Reviewer #1: Review comments

Comments to the Author

 “1. The abstract can be further refined. For example, the research results in the abstract can be summarized into three points.”

Response

Thanks for the valuable comment, we have re-summarized the research results in the abstract into three points. The new Abstract are as follows, the added and modified parts are marked in red:

Abstract: The impact of Outward Foreign Direct Investment (OFDI) on the global value chain (GVC) position of the manufacturing industry may vary significantly depending on the trade modes. This paper analyzes the impact and mechanism of outward FDI on the GVC position of China's manufacturing industry from the perspective of a typical binary trade structure, based on the calculation of the GVC position of China's manufacturing industry under general trade and processing trade from 2003 to 2018.The result shows that: (1) The GVC position of China's processing trade manufacturing industry is far lower than that of the general trade manufacturing industry. Specifically, from 2003 to 2018, the GVC position of China's general trade manufacturing industry increased from 2.76 to 2.90, while the GVC position of the processing trade manufacturing industry remained around 1.93. (2) Outward FDI enhances the GVC position of the general trade manufacturing industry through reverse technology spillover effects, industry structure upgrading effects, and export scale expansion effects. (3) Outward FDI has an inhibitory effect on the ascent of the processing trade manufacturing industry in the GVC. The research conclusion provides theoretical support for China to formulate outward FDI strategies based on different trade modes under the new pattern of dual circulation, to achieve the transformation and upgrading of the manufacturing industry and the ascent of the GVC.

“2. The title of Figure 1 is too long, you should make it shorter, also for Figure 2.”

Response

Thanks for this instructive comment, we have shortened the title of Figure 1 and Figure 2. The new title are as follows, the modified parts are marked in red:

Figure1 GVC position of China's manufacturing industry in different trade modes. Data source: drawn based on the calculated upstream index of China's manufacturing GVC.

（a）The overall GVC position distribution dynamics of Chinese manufacturing industry

（b）The GVC position distribution dynamics of general trade manufacturing industry

（c）The GVC position distribution dynamics of processing trade manufacturing industry 

Figure2 Dynamic evolution of GVC position of different trade modes in manufacturing industry. Data source: drawn based on the calculated upstream index of China's manufacturing GVC.

“3. It is suggested to further improve the innovation points.”

Response

Thanks for this instructive comment, this suggestion makes me feel that there are some problems in the innovation points. In the process of revising the manuscript, we have reorganized, summarized and enriched the innovation points of this paper. The improved innovation points are shown as follows:

Compared with previous research, this paper has three innovations: First, based on the large proportion of processing trade in China, this paper uses the world input-output table data which distinguishes between general trade and processing trade, to calculate the GVC position of manufacturing under different trade modes. It will further refine the relevant research on China's GVC. Second, the large proportion of processing trade may have an impact on the general conclusion that OFDI promotes the position of GVC. Therefore, different from the perspective of previous research, this paper examines the impact of OFDI on the GVC position of the general trade manufacturing industry and the processing trade manufacturing industry based on the perspective of China's typical dualistic trade structure. Third, considering the differences between general trade and processing trade in terms of production efficiency and technology absorption capacity, this paper explores the mechanism of OFDI's impact on the GVC position of manufacturing under different trade modes from the aspects of reverse technology spillover, industry structure, and export scale.

 “4. Why you divide the industries to 2 kinds? Medium and low tech industries, High tech industry, why not 3 kinds?”

Response

Thanks for the insightful question. We will respond from the following two points, and we hope that this response appropriately answers your question:

First, to highlight the difference between general trade manufacturing industry and processing trade manufacturing industry. In contrast to the general trade manufacturing industry, the processing trade manufacturing industry has low technology content and low added value, which may be the main reason why the processing trade manufacturing industry's GVC position is lower than that of the general trade manufacturing industry. As a result, the aforementioned hypothesis is evaluated by categorizing the industry into medium and low technology industries, as well as high technology industries.

Second, it emphasizes the impact of technical innovation level on the industry's global value chain position. Highlighting the high-tech industry from all industries can represent whether technological innovation is an essential element influencing the GVC position, mirroring the mechanism analysis in the article.

 “5. Some fresh papers can be added into references, eg:

Liu, F., Sim, J. Y., Sun, H., Edziah, B. K., Adom, P. K., & Song, S. (2023). Assessing the role of economic globalization on energy efficiency: Evidence from a global perspective. China Economic Review, 77, 101897. https://doi.org/10.1016/j.chieco.2022.101897.”

Response

Thanks for this instructive comment. According to the referee’s instructive comments, we have added some newest references of this study, and the added references are marked in red in the revised manuscript. Thanks for your instructive comment sincerely. The newest reference added are as follows:

6. Liu F, Sim J Y, Sun H, Edziah B K, Adom P K, Song S. Assessing the Role of Economic Globalization on Energy Efficiency: Evidence from A Global Perspective. China Economic Review, 2003;77, 101897. https://doi.org/10.1016/j.chieco.2022.101897

7. Yue W. Foreign Direct Investment and the Innovation Performance of Local Enterprises. Humanities and Social Sciences Communications,2022; 9, 252. https://doi.org/10.1057/s41599-022-01274-6

8. Zhang W, Zhang S, Chen F, Wang Y, Zhang Y.Does Chinese Companies' OFDI Enhance Their Own Green Technology Innovation?. Finance Research Letters,2023;56,104113. https://doi.org/10.1016/j.frl.2023.104113

23. Li X, Zhou W, Hou J. Research on The Impact of OFDI on The Home Country's Global Value Chain Upgrading. International Review of Financial Analysis, 2021;77,101862. https://doi.org/10.1016/j.irfa.2021.101862

50. Amendolagine V, Presbitero A F, Rabellotti R, Sanfilippo M. Local Sourcing in Developing Countries: The Role of Foreign Direct Investments and Global Value Chains. World Development,2019;113,73-88. https://doi.org/10.1016/j.worlddev.2018.08.010

51. Adarov A, Stehrer R. Implications of Foreign Direct Investment, Capital Formation and Its Structure for Global Value Chains. The World Economy,2021;44(11),3246-3299. https://doi.org/10.1111/twec.13160

52. Fernande A M, Kee H L, Winkler D. Determinants of Global Value Chain Participation: Cross-Country Evidence. The World Bank Economic Review,2022; 36(2),329-360. https://doi.org/10.1093/wber/lhab017

53. Hsu W T, Lu Y, Luo X, Zhu L M. Foreign Direct Investment and Industrial Agglomeration: Evidence from China. Journal of Comparative Economics,2023;5(2),610-639. https://doi.org/10.1016/j.jce.2022.12.004

54. Thompson P, Zang W Y. The Relationship Between Foreign Direct Investment and Domestic Entrepreneurship: The Impact and Scale of Investments in China. Growth and Change, 2023;1-42. https://doi.org/10.1111/grow.12671

Comments from the reviewers

Reviewer #2: Review comments

Comments to the Author

“1. Thank you for your interest to contribute into the PLOS journal.”

Response

We sincerely thank the referee’s interest and affirmation to our research, and we are also grateful for the referee’s instructive comments. 

“2. Main idea and topic.

The research idea is interesting indeed. However, the title needs to showcase the elements of general trade and processing trade. Thus, the title need improvements.”

Response

Thanks for this instructive comment. According to the referee’s instructive comments, we have modified the title into “Outward Foreign Direct Investment and GVC Position of Manufacturing Industry: A Perspective on China's general trade and processing trade structure”. In order to emphasize the elements of general trade and processing trade.

“3. Abstract.

There is a lack on explaining the results which only shows that the GVC position of China's processing trade manufacturing 11 industry is far lower than that of the general trade manufacturing industry. A more specific is needed to highlight any figures or trends that revealed such results.”

Response

Thanks for the valuable comment, we have re-summarized the research results in the abstract into three points. The new Abstract are as follows, the added and modified parts are marked in red:

Abstract: The impact of Outward Foreign Direct Investment (OFDI) on the global value chain (GVC) position of the manufacturing industry may vary significantly depending on the trade modes. This paper analyzes the impact and mechanism of outward FDI on the GVC position of China's manufacturing industry from the perspective of a typical binary trade structure, based on the calculation of the GVC position of China's manufacturing industry under general trade and processing trade from 2003 to 2018.The result shows that: (1) The GVC position of China's processing trade manufacturing industry is far lower than that of the general trade manufacturing industry. Specifically, from 2003 to 2018, the GVC position of China's general trade manufacturing industry increased from 2.76 to 2.90, while the GVC position of the processing trade manufacturing industry remained around 1.93. (2) Outward FDI enhances the GVC position of the general trade manufacturing industry through reverse technology spillover effects, industry structure upgrading effects, and export scale expansion effects. (3) Outward FDI has an inhibitory effect on the ascent of the processing trade manufacturing industry in the GVC. The research conclusion provides theoretical support for China to formulate outward FDI strategies based on different trade modes under the new pattern of dual circulation, to achieve the transformation and upgrading of the manufacturing industry and the ascent of the GVC.

“4. Introduction and literature review.

a) In this paper, it has a strong background and clear direction. Furthermore its objectives are justified clearly.

b) However, some important elements that have been highlighted in the paper were not been stressed in the introduction such as the elements of reverse technology spillover, structure upgrading and export scale expansion.

c) Why the paper used the binary trade structure?

d) Elaborate further the world input-output model. How this model is in line to extract that results in this study?”

Response

For comment a): We would like to express our gratitude to the referee for the recognition of the research background and research objectives of this paper.

For comment b): Thanks for this instructive comment. According to the referee’s instructive comments, we have improved the introduction section to stress the elements of reverse technology spillover, structure upgrading and export scale expansion. The new introduction are as follows, the added and modified parts are marked in red:

Introduction

The role of global value chain (GVC) participation not only affects a country's ability to benefit from international trade but also serves as a key determinant for its industrial transformation and upgrading [1-2]. China's manufacturing sector has long been a low- to mid-end supplier to developed countries, running the risk of "low-end lock-in" despite actively integrating into the GVC through low value-added and low-tech processing and production activities[3-5]. How to improve China's GVC position has become an important issue in accelerating the transformation and upgrading of China's manufacturing sector under the new development pattern of the dual circulation.

Out foreign direct investment (OFDI) as an important component of the dual cycle, has the hub function of connecting domestic and foreign markets. The rapid development of OFDI not only help China's manufacturing industry to integrate foreign capital, labor and other resources, optimize the allocation of domestic resources [6], but also help China's manufacturing industry to get close to foreign R&D and design links [7]. So that it is easier to obtain information such as key production technologies and advanced management experience, also to promote its scientific and technological innovation and value creation ability [8]. China's OFDI topped that of the United States for the first time in 2020, ranking first globally. In 2021, China's OFDI reached $145.19 billion, a year-on-year increase of 9.2% [9]. Thus, China has become an important player in global OFDI.

It is worth noting that, China's manufacturing industry is large in scale and has a typical dualistic trade structure [10-11], with notable distinctions between general trade and processing trade in terms of technological content and benefits [12]. General trade manufacturing has a more complicated production process, but it also has stronger production efficiency, technology absorption capabilities, and global competitiveness. While processing trade manufacturing has low production technology content and trade value-added, it does have the qualities of "importing large quantities of raw materials and exporting finished products" [13-15]. This not only creates a large disparity in the GVC position between China's general trade and processing trade manufacturing but may also result in significant differences in the impact and effectiveness of OFDI on the GVC position of different trade types. Clarifying the effect and mechanism of OFDI on the GVC position of general trade and manufacturing processing trade can therefore serve as a guide for China's OFDI strategy under the new development pattern of dual circulation, promoting China's ascent in the GVC and realizing the transformation and upgrading of its manufacturing industry.

As economic globalization has advanced, more academics have concentrated on the effect of OFDI on the GVC. Based on their findings, pertinent studies can be split into two categories. On the one hand, some scholars believe that OFDI can enhance a country's GVC position both in the home and host countries [16-19]. For example, Yang & Luo found that OFDI has raised China's GVC position through reverse technology spillovers[20], While Liu et al.discovered that OFDI has helped China ascend in the GVC by promoting product quality and functional upgrading[21]. Nie & Li indicated that the deindustrialization of the service sector has a significant positive effect on improving a firm's GVC position through OFDI[22]. The above related literature mainly reflects that OFDI has technology spillover effect, which further improves the GVC position. The theory of technology localization and the theory of technological innovation industrial upgrading provide theoretical support for the technology spillover effect of OFDI. In addition, the impact mechanism of OFDI on a country's GVC position also includes marginal industry transfer effects [23], export creation effects [24], market competition effects [25] and so on. 

On the one hand, some scholars argue that OFDI may have neutral or negative effects on a country's GVC position [26]. This is mainly because OFDI may have substitution effects on domestic investment [27-29], leading to the crowding out of resources needed for domestic industry development,such as research and development investment. Additionally, OFDI aimed at marginal industry transfer may carry the risk of industrial hollowing-out [30-31].

Unlike most countries, China has a relatively high proportion of processing trade in its foreign trade, which has gradually attracted the attention of the academic community. Processing trade enterprises generally have lower productivity and international competitiveness due to their low technological content and value-added features [32]. Therefore, China's processing trade and general trade exhibit significant differences in terms of trade transformation path [33], manufacturing service level [34], and production chain position. Wang et al. pointed out that processing trade is closer to downstream consumption in the production chain, and if general trade and processing trade are not distinguished in the measurement process, it may distort China's true GVC position[35]. To this end, Ma & Li and Peng & Wu further described the upstream index of China's manufacturing industry from the perspectives of general trade and processing trade. They found that China's GVC position of general trade was significantly higher than that of processing trade.[36-37]

Reviewing the relevant literature, it is found that most studies on the impact of OFDI on a country's GVC position view China's manufacturing industry as a whole, with few studies examining the impact of OFDI on China's manufacturing industry GVC position from the perspective of general trade and processing trade. Due to the significant differences between general trade and processing trade in terms of manufacturing efficiency, technology absorption ability, and international competitiveness, the impact of OFDI on the GVC position of general trade and processing trade manufacturing may differ, which has been overlooked in previous literature. Therefore, this paper uses national and regional input-output table data that differentiate between different trade modes to measure China's GVC position in general trade and processing trade manufacturing. Based on this, the paper examines the impact and mechanism of OFDI on the GVC position of manufacturing under different trade modes.

Compared with previous research, this paper has three innovations: First, based on the large proportion of processing trade in China, this paper uses the world input-output table data which distinguishes between general trade and processing trade, to calculate the GVC position of manufacturing under different trade modes. It will further refine the relevant research on China's GVC. Second, the large proportion of processing trade may have an impact on the general conclusion that OFDI promotes the position of GVC. Therefore, different from the perspective of previous research, this paper examines the impact of OFDI on the GVC position of the general trade manufacturing industry and the processing trade manufacturing industry based on the perspective of China's typical dualistic trade structure. Third, considering the differences between general trade and processing trade in terms of production efficiency and technology absorption capacity, this paper explores the mechanism of OFDI's impact on the GVC position of manufacturing under different trade modes from the aspects of reverse technology spillover, industry structure, and export scale.

For comment c): Thanks for this valuable question, we will quote the third paragraph in the introduction for explanation, and we hope that our response appropriately answers your query: 

China's manufacturing industry is large in scale and has a typical dualistic trade structure [10-11], with notable distinctions between general trade and processing trade in terms of technological content and benefits [12]. General trade manufacturing has a more complicated production process, but it also has stronger production efficiency, technology absorption capabilities, and global competitiveness. While processing trade manufacturing has low production technology content and trade value-added, it does have the qualities of "importing large quantities of raw materials and exporting finished products" [13-15]. This not only creates a large disparity in the GVC position between China's general trade and processing trade manufacturing but may also result in significant differences in the impact and effectiveness of OFDI on the GVC position of different trade types. Clarifying the effect and mechanism of OFDI on the GVC position of general trade and manufacturing processing trade can therefore serve as a guide for China's OFDI strategy under the new development pattern of dual circulation, promoting China's ascent in the GVC and realizing the transformation and upgrading of its manufacturing industry.

For comment d): Thanks for this instructive comment. According to the referee’s instructive comments, we have elaborated further the world input-output model. The model is explained as follows, the added parts are marked in red:

Measurement model for the position of China's manufacturing industry in the GVC

According to the ICIO model, a world input-output table is constructed that distinguishes between general trade (G) and processing trade (P) activities in China (as shown in Table 1). It is assumed that there are (N+1) countries or regions in the world, and China is divided into two "regions" that engage in either general trade or processing trade. Each country or region has M industries, of which Q are manufacturing industries that produce specific products or services. It is worth noting that processing trade enterprises can only engage in production, processing, and export businesses, so processing trade cannot provide intermediate consumption goods for general trade, nor can it provide final consumption goods for China as a whole.

In Table 1, the superscripts G and P represent China's general trade and processing trade, respectively. Z is an (N+1) * (N+1) matrix representing the intermediate input and usage of goods among countries, Y and X are column vectors of (N+1) * 1, representing a country's final use (including household final consumption, non-profit institutions serving households, government final consumption, gross fixed capital formation, changes in inventories and valuables, and foreign purchases) and total output, respectively. VA is a row vector of 1 * (N+1), representing the value added that each country obtains in production. The superscript "‘" indicates a transpose operation.

Table 1. World input-output table distinguishing China's general trade from its processing trade

 Intermediate using Final using Total out-put

 China (C) Country1 … Country N China

(C) Country1 … Country N 

 Processing trade General trade 

Inter-mediate using C Process-ing trade 0 0 ZP1 … ZPN 0 YP1 … YPN XP

 General trade ZGP ZGG ZG1 … ZGN YGC YG1 … YGN XG

 Country 1 Z1P Z1G Z11 … Z1N Y1C Y11 … Y1N X1

 … … … … … … … … … … …

 Country N ZNP ZNG ZN1 … ZNN YNC YN1 … YNN XN

Value added VAP VAG VA1 … VAN — — — — —

Total input (XP)' (XG)' (X1)' … (XN)' — — — — —

From the perspective of usage, when the market is cleared, the following equilibrium equation exists for Table 1:

 (1)

Assuming the direct input coefficient , where is the inverse matrix of the diagonal matrix of N country's total output vector. Equation (1) can be expressed as:

 (2)

Let , , andrespectively represent the world direct input coefficient matrix, total output column vector, and final use column vector. Then Equation (2) can be written as:

 (3)

In Equation (3), is the Leontief inverse matrix. From Formula (3), the form of the infinite sequence of Formula (4) can be obtained:

 (4)

The number of stages from the first item in Formula (4) to the final demand is 1, which is actually the number of production stages of the final product. The number of stages from the second term to final demand is 2:1 intermediate goods production stage and 1 final goods production stage. Next, borrowing the research method of Alfaro & Chor , assuming that the distance between any two production stages in a country is equal and is 1[46]. Then the output of the country industry can be expressed as:

 (5)

In the above equation, x represents total output, a represents the coefficient of input demand, and y represents total input. Continue to multiply the right-hand side of Equation (5) by the sum of the distances from their corresponding final consumption expenditures plus one and then divide by output. The weighted average position of a single industry in the production chain is thus calculated. The output upstream index of industry r in country i is obtained, as shown in Formula (6):

 (6)

Based on the upstream degree index of the industry output of country calculated in Equation (6), its position in the GVC can be determined. The larger the value , the greater the proportion of intermediate product output supplied by the industry to other industries in its total output, and the more closely it is connected with other industries. In other words, the larger the value , the closer the industry in country is to the production side of the production-consumption chain, which further makes its position in the GVC higher. Conversely, the smaller the value , the closer the industry in country is to the final consumption side, and the lower its position in the GVC division of labor.

Furthermore, by simplifying Equation (6) and representing it in matrix form, we have:

 (7)

Based on Equation (6), multiplying the upstream degree index of the manufacturing industries in country by the proportion of each industry in the total output of the manufacturing industry and then adding them up, we can obtain the upstream degree index of the overall manufacturing industry in country , as shown in Equation (8). It should be noted that this study assumes that China's general trade and processing trade are two "economic entities". Therefore, the upstream degree index of China's general trade and processing trade calculated using Equation (8) does not represent the GVC position of China's overall manufacturing industry. Therefore, when calculating the upstream degree index of China's overall manufacturing industry, the general trade and processing trade of China in Table 1 should be merged first before calculating.

 (8)

“5. Research methodology / results / Data source / analysis.

a) Author did well in explaining the analysis and measure to extract the data particularly the general trade andprocessing trade GVC.

b) In this paper, it has a strong background and clear direction. Furthermore its objectives are justified clearly.

c) The author needs source on a statement highlighted about ‘China Science and Technology Statistical Yearbook’.

d) As for the mechanism analysis, author needs to add evidences of similarities or differences from this analysis against previous studies.

e) Interestingly, processing manufacturing seems to not change significantly. Explain.”

Response

For comment a): Thanks a lot for the referee’s evaluation of the analysis and measure to extract the data of this paper.

For comment b): We would like to express our gratitude to the referee for the recognition of the research background and research objectives of this paper.

For comment c): Thanks for the valuable comment. During the revision of this paper, we have added footnote on page 12 of the paper, to emphasize the source and explanation of the data. The added footnote is as follow:

The source of "China Science and Technology Statistical Yearbook": https://navi.cnki.net/knavi/yearbooks/YBVCX/detail?uniplatform=NZKPT&language=chs. The expenditure on research and development, the number of patent applications, the total number of scientific and technological staff, the total number of employees, and other statistics are used in this paper, and the appropriate data have been submitted to the PLOS ONE system.

For comment d): Thanks for this instructive comment. According to the referee’s instructive comments, we have added some newest references of similarities or differences from this analysis against previous studies in the mechanism analysis section. The added references are marked in red:

Mechanism Analysis

As seen by the previous regression results, the influence of OFDI on the manufacturing industry's GVC position varies dramatically between trade modes. A mechanism analysis is performed from the perspectives of reverse technology spillovers, industry structure, and export scale to further investigate the reasons for these disparities. The mechanism analysis is carried out in two stages. The basic explanatory variables are regressed against patent applications, industry structure indicators, and export scale in the first stage, and the results reveal that OFDI can produce reverse technology spillover effects, industry structure upgrading effects, and export scale expansion effects. This conclusion is similar with the findings of Yang & Luo , Yue and Zhang et al. [7,8,20]et al. The second stage is to regress the mechanism variables' OFDI and interaction terms against the manufacturing industry's GVC position. Table 8 displays the results.

The findings of assessing the reverse technology spillover effects, industry structure upgrading effects, and export scale expansion impacts of OFDI on the GVC position of the manufacturing industry are shown in columns (1)-(3), (4)-(6), and (7)-(9). Except for the negligible interaction factors in columns (3), (6), and (9), it can be seen that all other interaction variables are significant. This suggests that while OFDI can improve the overall and general trade manufacturing GVC position through reverse technology spillover effects, industry structure upgrading effects, and export scale expansion effects, its effect on processing trade manufacturing is negligible, limiting its upward mobility in the GVC. Hypothesis 2 is confirmed. This conclusion not only highlights the difference between general trade and processing trade in terms of technical content [11,15,32], indirectly supports previous research on the promotion of a country's or region's GVC position by OFDI [19,50,52], but also broadens the research boundary of OFDI and GVC, allowing for further refinement of the research.

Based on this, the reason why OFDI suppresses the upward mobility of processing trade manufacturing in the GVC may be that OFDI of processing trade enterprises reduces the parent company's market share in overseas markets and exacerbates homogenized competition, which is not conducive to the optimization of resource allocation such as R&D costs within the enterprise [53-54], and thus not conducive to the improvement of its GVC position.

For comment e): Thanks for this valuable question. In response to the referee's questions, we answer from the three points listed below:

First, processing trade refers to economic activities in which producers buy a large number of intermediate inputs from other countries and then export final products after simple processing or assembly. As a result, it is simple to get into "low-end lock" in GVC.

Second, the production efficiency, technology content and technology absorption and conversion capacity of processing trade manufacturing industry are generally low, which makes it difficult for processing trade enterprises to obtain foreign advanced technology and other resources through OFDI and transfer related industries. Furthermore, because processing trade firms have little export value-added content, they have minimal potential for lowering production and trade costs through OFDI.

Third, China's processing trade manufacturing industry is near the bottom of the GVC. At the same time, developed countries continue to control the processing and assembly of precise equipment. To protect their own interests and their leading position in GVC, industrialized countries have erected many technological, personnel flow, and market barriers to foreign investment that are not conducive to processing trade firms climbing GVC through OFDI.

“6. Discussion and conclusion

Conclusion is found to weak and has less aspect of generalization on the impact and contribution of the results in particular to China and the industry. Further a more comprehensive view of the study are needed to conquer the overall picture.

Conclusion needs to add the overall results and unique findings that can contribute to the knowledge area.”

Response

Thanks for the valuable comment, this suggestion makes me feel that there are some problems in the conclusion section. In the process of revision, we enrich the conclusion of this paper, and re-analyze the theoretical value and practical significance of the conclusion. In addition, at the end of the paper, we add the shortcomings of this study. The added and modified parts are marked in red:

Conclusion

This paper examines the impact and mechanism of OFDI on the GVC position of China's manufacturing industry from the perspectives of general trade and processing trade, based on the calculation of the GVC position of China's manufacturing industry as a whole and its trade mode, and draws the following conclusions: (1) Processing trade manufacturing has a substantially lower GVC position in China than general trade. From 2003 to 2018, China's general trade manufacturing industry's GVC position climbed from 2.76 to 2.90, while the processing trade manufacturing industry's GVC position remained around 1.93. (2) OFDI can greatly increase general trade manufacturing's GVC position while impeding the upward mobility of processing trade manufacturing in the GVC. This conclusion remains after a variety of robustness tests and consideration of endogeneity issues. (3) Mechanism analysis reveals that OFDI promotes the global value chain status of the general trade manufacturing industry via the reverse technology spillover effect, industrial structure upgrading effect, and export scale expansion effect, but the above effect does not appear for the processing trade manufacturing industry. Perhaps it is also a significant cause of conclusion (2).

According to the research findings, the promotion effect of OFDI on the global value chain status of a country's or region's manufacturing sector is primarily concentrated in the general trade manufacturing industry, rather than the processing trade manufacturing business. As a result, when a country, such as China, has a high proportion of processing trade, it must evaluate the influence of diverse trade modes while evaluating issues related to its global value chain. Similarly, when analyzing the impact of OFDI on global value chain status, we need differentiate between regular commerce and processing trade. Our research not only refines relevant research on FDI and the global value chain, but also provides a reference for China to create FDI strategy under the new pattern of twofold circulation, in order to climb the global value chain and realize manufacturing sector transformation and upgrading.

One of the important limitations of the study is that we could not find the reason why OFDI inhibits China's processing trade manufacturing industry from climbing the global value chain. we left this question for future research.

Finally, the reference section is rewritten in accordance with the foregoing change, and unsuitable expressions throughout the article are revised.

Furthermore, we improve the article format, such as the reference and title formats, such as the reference and title formats. The modification of this issue was not marked in the revised draft.

If editors or reviewers have further comments, please inform us in time, and the authors will make improvements as soon as possible as required.

---

## [Decision Letter · Decision Letter 1]

2 Oct 2023

PONE-D-23-19895R1Outward Foreign Direct Investment and GVC Position of Manufacturing Industry: A Perspective on China's General Trade and Processing Trade StructurePLOS ONE

Dear Dr. Hu,

Thank you for submitting your manuscript to PLOS ONE. After careful consideration, we feel that it has merit but does not fully meet PLOS ONE’s publication criteria as it currently stands. Therefore, we invite you to submit a revised version of the manuscript that addresses the points raised during the review process.

We look forward to receiving your revised manuscript.

Kind regards,

Shujahat Haider Hashmi, PhD Regional Economics

Academic Editor

PLOS ONE

Journal Requirements:

**Additional Editor Comments:**

The reviewers have accepted your revisions but there are still some language and structural issues; some sentences are too long and absurd. Kindly edit the manuscript thoroughly and improve its quality on serious note.

Reviewers' comments:

Reviewer's Responses to Questions

**Comments to the Author**

1. If the authors have adequately addressed your comments raised in a previous round of review and you feel that this manuscript is now acceptable for publication, you may indicate that here to bypass the “Comments to the Author” section, enter your conflict of interest statement in the “Confidential to Editor” section, and submit your "Accept" recommendation.

Reviewer #1: All comments have been addressed

Reviewer #2: All comments have been addressed

2. Is the manuscript technically sound, and do the data support the conclusions?

Reviewer #1: Yes

Reviewer #2: Yes

3. Has the statistical analysis been performed appropriately and rigorously? 

Reviewer #1: Yes

Reviewer #2: Yes

4. Have the authors made all data underlying the findings in their manuscript fully available?

Reviewer #1: Yes

Reviewer #2: Yes

5. Is the manuscript presented in an intelligible fashion and written in standard English?

Reviewer #1: Yes

Reviewer #2: Yes

6. Review Comments to the Author

Reviewer #1: Now all ok for the structure and discussions, only one point:

Some sentences are too long, so it is better to make it shorter, eg the following is too long:

This paper analyzes the impact and mechanism of outward FDI on the GVC position of China's manufacturing industry from the perspective of a typical binary trade structure, based on the calculation of the GVC position of China's manufacturingindustry under general trade and processing trade from 2003 to 2018.

Reviewer #2: Outward Foreign Direct Investment and GVC Position of Manufacturing Industry: A Perspective on China's General Trade and Processing Trade Structure

1) Thank you for your response and improvements.

Abstract:

There is a lack on explaining the results which only shows that the GVC position of China's processing trade manufacturing 11 industry is far lower than that of the general trade manufacturing industry. A more specific is needed to highlight any figures or trends that revealed such results.”

The author have added specific and improved the abstract with the trend and additional information.

2) Introduction and literature review

However, some important elements that have been highlighted in the paper were not been stressed in the introduction such as the elements of reverse technology spillover, structure upgrading and export scale expansion.

Author improved and make justification to several elements: reverse technology spillover, structure upgrading and export scale expansion.

Why the paper used the binary trade structure?

Author improved but still not justify as the link and explanation on specific binary trade still not clear

Elaborate further the world input-output model. How this model is in line to extract that results in this study?”

Author did extends the discussion on input-output model and how its contributed in the study.

3) Research methodology

• The author needs source on a statement highlighted about ‘China Science and Technology Statistical Yearbook’.

Author did additional explanation and have added footnote to emphasize the source and explanation of the data.

• As for the mechanism analysis, author needs to add evidences of similarities or differences from this analysis against previous studies.

Author has added the explanation

• Interestingly, processing manufacturing seems to have no changes significantly. Explain.”

Author did additional explanation with three perspectives which are acceptable.

4) Conclusion

Conclusion is found to be weak and has less aspect of generalization on the impact and contribution of the results in particular to China and the industry. Further a more comprehensive view of the study are needed to conquer the overall picture. Conclusion needs to add the overall results and unique findings that can contribute to the knowledge area.”

Since the author did the improvements in the conclusion particularly on the generalization and contribution thus the paper has met the standard

7. PLOS authors have the option to publish the peer review history of their article (what does this mean?). If published, this will include your full peer review and any attached files.

Reviewer #1: No

Reviewer #2: **Yes: **M H Jedin

---

## [Author Response · Author response to Decision Letter 1]

21 Oct 2023

Response to the Editor and Referee

Title: Outward Foreign Direct Investment and GVC Position of Manufacturing Industry: A Perspective on China's General Trade and Processing Trade Structure

Authors: Fei Ren, Dong Le, Ziyu Hu* 

Dear Prof :

We really appreciate your response concerning the submission of this paper to PLOS ONE. We also would like to express great appreciation to you and the reviewers for these constructive comments. We have thoughtfully taken into these comments. The explanations of what we have revised in response to the reviewers’ comments are given point by point. We hope that all these changes fulfill your requirements as well as the reviewer. Correspondence and calls about this paper should be directed to author at the following address and e-mail. Thanks very much again for your attention to our paper.

Once again, thank you for your time and help to our paper processing.

Yours Sincerely,

Ziyu Hu

Address: Economics School, Zhongnan University of Economics and Law, Wuhan 430073, China;

E-mail:994138246@qq.com

For your guidance, itemized responses to reviewers’ comments are appended below. 

1. Overall

Firstly, we are thankful to the reviewer for the highly instructive comments and valuable suggestions, which have been considered in this revision. Secondly, we rectified the grammatical inaccuracies included in the previous manuscript. Furthermore, certain sentences were simplified without compromising their original significance. Thirdly, we confirmed that there are no withdrawals in the references and adjusted the formatting of the references to conform to the journal requirements. Finally, we have added a title page in the manuscript to indicate author information, and referred to Table 4 in the main text of manuscript.

2. Detailed responses 

Thank you very much for reviewing the revised manuscript. We believe that your further comments and suggestions are highly constructive and very helpful for our paper. We have thoughtfully taken into these comments and responded to your constructive suggestions from point by point outlined below. We hope we have addressed all of your concerns.

For the sake of presentation, the comments of the reviewers are reproduced in italics, and our responses are given in plain. 

Comments from the reviewers

Reviewer #1: Review comments

Comments to the Author

 “1. Now all ok for the structure and discussions, only one point:Some sentences are too long, so it is better to make it shorter, eg the following is too long:

This paper analyzes the impact and mechanism of outward FDI on the GVC position of China's manufacturing industry from the perspective of a typical binary trade structure, based on the calculation of the GVC position of China's manufacturing industry under general trade and processing trade from 2003 to 2018.”

Response

We appreciate the insightful feedback you have provided. The grammatical faults present in the initial manuscript have been rectified, and certain sentences have been shortened to enhance the clarity and simplicity of this paper. The revised sentence we have highlighted in red in the original manuscript. For illustrative purposes, the modified abstract is presented hereafter:

Abstract: Depending on the trading modes, the effect of Outward Foreign Direct Investment (OFDI) on the manufacturing industry's position within the global value chain (GVC) may differ considerably. This paper examines the GVC position of China's manufacturing industry from 2003 to 2018, specifically focusing on the general trade and processing trade. Drawing upon this premise, this paper analyzes the effect and mechanism by which outward FDI influences the GVC position of China's manufacturing industry. The result demonstrates that: (1) China's processing trade manufacturing industry has a much lower GVC position than general trade manufacturing industry. The GVC position of China's general trade manufacturing industry rose from 2.76 to 2.90 from 2003 to 2018, while the processing trade manufacturing industry remained around 1.93. (2) Outward FDI boosts the GVC position of general trade manufacturing industry through facilitating reverse technology spillover, inducing industry structure upgrading, and enabling export scale expansion. (3) Outward FDI hinders the GVC position growth of processing trade manufacturing industry. The research findings offer theoretical backing for China to develop outward FDI strategies that are tailored to different trading modes within the new framework of dual circulation. These strategies aim to facilitate the transformation and advancement of the manufacturing industry, as well as the growth of the GVC position.

Comments from the reviewers

Reviewer #2: Review comments

Comments to the Author

“1. Thank you for your response and improvements.

Abstract

‘There is a lack on explaining the results which only shows that the GVC position of China's processing trade manufacturing 11 industry is far lower than that of the general trade manufacturing industry. A more specific is needed to highlight any figures or trends that revealed such results.’

The author have added specific and improved the abstract with the trend and additional information.”

Response

We sincerely thank the reviewer’s interest and affirmation to our research, and we are also grateful for the reviewer’s instructive comments. 

“2. Introduction and literature review

• ‘However, some important elements that have been highlighted in the paper were not been stressed in the introduction such as the elements of reverse technology spillover, structure upgrading and export scale expansion.’

Author improved and make justification to several elements: reverse technology spillover, structure upgrading and export scale expansion.

• ‘Why the paper used the binary trade structure?’

Author improved but still not justify as the link and explanation on specific binary trade still not clear

• ‘Elaborate further the world input-output model. How this model is in line to extract that results in this study?’

Author did extends the discussion on input-output model and how its contributed in the study.”

Response

Thank you very much to the reviewer for recognizing our last revision. To provide an explanation for our utilization of the binary trade structure, we conducted additional analysis on the distinction between general trade manufacturing and processing trade manufacturing in response to the second concern you raised. The relevant analysis is shown below:

It is noteworthy to mention that the manufacturing industry in China has a substantial size and is characterized by a dual trade structure [10-11]. The dual trade structure primarily encompasses two dominant modes of trade, namely general trade and processing trade. The above two have differences in specific connotation. General trade refers to enterprises in China with import and export operation rights, engaging in unilateral import or unilateral export trade activities. Processing trade refers to the operations of enterprises that engage in the importation (or exportation) of raw and auxiliary materials, parts and components, components, packaging materials, and other materials. These materials are then processed or assembled, resulting in the production of finished goods that are then re-exported (or re-imported). According to China Customs figures, the average proportion of China's processing trade exports between 2002 and 2007 was recorded at 53.07%. Despite experiencing a gradual decline in recent years, this proportion remains as high as 20.1% in 2022. China's import and export commerce exhibits a distinct dual trade structure.

In the realm of manufacturing, notable distinctions exist between general trade and processing trade, particularly with regard to technological sophistication and profitability [12]. The general trade manufacturing industry exhibits a more intricate production chain, hence enhancing its productivity, capacity to absorb technology, and worldwide competitiveness. Conversely, the processing trade manufacturing industry is distinguished by the principles of "both ends out" and "large import, large export". The phrase "both ends out" refers to the scenario when the procurement of raw materials and the distribution of finished products occur in the global market. "Large import" refers to the raw materials required for the production of enterprises, mainly dependent on imports. "Large exports" means that enterprises' large-scale exports are based on large imports. It is evident that the aforementioned features pertaining to the processing trade manufacturing industry contribute to a diminished technological content in production and value added in trade [13-15]. 

Based on our analysis, it can be inferred that the differences between general trade manufacturing and processing trade manufacturing potentially contributes to a more pronounced divergence in the China's GVC position. This could also cause huge variances in OFDI's impact on China's manufacturing industry's GVC position under different trading forms. Therefore, understanding how OFDI affects the GVC position of general trade manufacturing industry and processing trade manufacturing industry would help China restructure and grow its manufacturing sector industry.

“3. Research methodology

• ‘The author needs source on a statement highlighted about ‘China Science and Technology Statistical Yearbook’.’

Author did additional explanation and have added footnote to emphasize the source and explanation of the data.

• ‘As for the mechanism analysis, author needs to add evidences of similarities or differences from this analysis against previous studies.’

Author has added the explanation

• ‘Interestingly, processing manufacturing seems to have no changes significantly. Explain.’

Author did additional explanation with three perspectives which are acceptable.”

Response

We would like to thank the reviewer for the comments and evaluations. We appreciate your acknowledgement of the revised paper.

“4. Conclusion

‘Conclusion is found to be weak and has less aspect of generalization on the impact and contribution of the results in particular to China and the industry. Further a more comprehensive view of the study are needed to conquer the overall picture. Conclusion needs to add the overall results and unique findings that can contribute to the knowledge area.

Since the author did the improvements in the conclusion particularly on the generalization and contribution thus the paper has met the standard.”

Response

We are deeply delighted that the revisions we submitted to this paper were approved by the reviewer. We express sincere gratitude for the favorable feedback provided by the reviewer.

If editor or reviewers have further comments, please inform us in time, and the authors will make improvements as soon as possible as required.

---

## [Decision Letter · Decision Letter 2]

4 Dec 2023

Outward Foreign Direct Investment and GVC Position of Manufacturing Industry: A Perspective on China's General Trade and Processing Trade Structure

PONE-D-23-19895R2

Dear Dr. Hu,

We’re pleased to inform you that your manuscript has been judged scientifically suitable for publication and will be formally accepted for publication once it meets all outstanding technical requirements.

Kind regards,

Shujahat Haider Hashmi, PhD Regional Economics

Academic Editor

PLOS ONE

Additional Editor Comments (optional):

Reviewers' comments:

Reviewer's Responses to Questions

**Comments to the Author**

1. If the authors have adequately addressed your comments raised in a previous round of review and you feel that this manuscript is now acceptable for publication, you may indicate that here to bypass the “Comments to the Author” section, enter your conflict of interest statement in the “Confidential to Editor” section, and submit your "Accept" recommendation.

Reviewer #1: All comments have been addressed

Reviewer #2: All comments have been addressed

2. Is the manuscript technically sound, and do the data support the conclusions?

Reviewer #1: Yes

Reviewer #2: Yes

3. Has the statistical analysis been performed appropriately and rigorously? 

Reviewer #1: Yes

Reviewer #2: Yes

4. Have the authors made all data underlying the findings in their manuscript fully available?

Reviewer #1: Yes

Reviewer #2: Yes

5. Is the manuscript presented in an intelligible fashion and written in standard English?

Reviewer #1: Yes

Reviewer #2: Yes

6. Review Comments to the Author

Reviewer #1: Now all points are addressed already, and the current revisions are very serious and meticulous, reflecting the author's rigorous attitude. So I suggest accepting publication.

Reviewer #2: Dear Author,

Abstract:

Authors did improved the abstract as suggested.

Introduction:

Author have improved the introduction section by emphasizing few elements such as reverse technology spillover, structure upgrading and export scale expansion.

The remaining statements and references also have been updated.

7. PLOS authors have the option to publish the peer review history of their article (what does this mean?). If published, this will include your full peer review and any attached files.

Reviewer #1: No

Reviewer #2: No

---

## [Editor Report · Acceptance letter]

7 Dec 2023

PONE-D-23-19895R2 

Outward Foreign Direct Investment and GVC Position of Manufacturing Industry: A Perspective on China's General Trade and Processing Trade Structure 

Dear Dr. Hu:

I'm pleased to inform you that your manuscript has been deemed suitable for publication in PLOS ONE. Congratulations! Your manuscript is now with our production department. 

Kind regards, 

on behalf of

Dr. Shujahat Haider Hashmi 

Academic Editor

PLOS ONE